



# Modelling of the shallow water table at high spatial resolution using Random Forests.

Julian Koch[1], Helen Berger[2], Hans Jørgen Henriksen[1], Torben Obel Sonnenborg[1]

[1]Department of Hydrology, Geological Survey of Denmark and Greenland (GEUS), Copenhagen, 1350, Denmark
[2]COWI A/S, Lyngby, 2800, Denmark

*Correspondence to*: Julian Koch (juko@geus.dk)

**Abstract.** Machine learning provides a great potential to model hydrological variables at a spatial resolution beyond the capabilities of traditional physically-based modelling. This study features an application of Random Forests (RF) to model the depth to the shallow water table, for a wintertime minimum event, at 50 m resolution over a 15,000 km$^2$ large domain in
Denmark. In Denmark, the shallow groundwater poses severe risks of groundwater induced flood events affecting both, urban and agricultural areas. The risk is especially critical in wintertime, when the shallow groundwater is close to terrain. In order to advance modelling capabilities of the shallow groundwater system and to provide estimates at scales required for decision making, this study introduces a simple method to unify RF and physically-based modelling. Results from the national water resources model in Denmark (DK-model) at 500 m resolution are employed as covariate in the RF model. Thereby, RF ensures
physical consistency at coarse scale and fully exhausts high-resolution information from readily available environmental variables. The vertical distance to the nearest waterbody was rated the most important covariate in the trained RF model followed by the DK-model. The validation test of the trained RF model was very satisfying with a mean absolute error of 79 cm and a coefficient of determination of 0.55. The resulting map underlines the severity of groundwater flooding risk in Denmark, as the average depth to the shallow groundwater is 1.9 m and approximately 29 % of the area is characterised with
a depth less than 1 m during a typical wintertime minimum event. This study brings forward a novel method to assess the spatial patterns of covariate importance of the RF predictions which contributes to an increased interpretability of the RF model. Quantifying uncertainty of RF models is still rare for hydrological applications. Two approaches, namely Random Forests Regression Kriging (RFRK) and Quantile Regression Forests (QRF) were tested to estimate uncertainties related to the predicted groundwater levels. This study argues that the uncertainty sources captured by RFRK and QRF can be considered
independent and hence, they can be combined to a total variance through simple uncertainty propagation.

## 1 Introduction

The shallow groundwater, defined as the uppermost water table, is a key state variable of the hydrological cycle having a wide range of vital implications on human health, terrestrial ecosystems, food security and energy production (Gleeson et al., 2016) . Following Fan et al. (2013), up to one third of the global land area is affected by the shallow groundwater, being either





directly groundwater-fed or having the water table or capillary fringe within plant rooting depths. In many regions of the world, groundwater aquifers are being depleted extensively by unsustainable anthropogenic activities (Richey et al., 2015). In addition, climate change affects groundwater recharge and storage which, in many cases exacerbate the resilience of shallow groundwater systems (Ferguson and Maxwell, 2010; Rodell et al., 2018).

There exists a broad relevancy of the shallow groundwater which expands beyond the hydrological science. For instance, Kahlown et al. (2005) and Zipper et al. (2015) studied the dependency between crop yield and the water table. They concluded that, in many agricultural settings, the groundwater played an essential role in meeting the crop water requirements. However, water tables too close to the surface resulted in reduced yields and both studies identified an optimal water table between 1 and 2 m below surface. Moreover, several studies highlighted the controlling mechanisms that the water table has on the energy

partitioning at the land surface inferring a link to the latent heat flux and the delineation of water- and energy-limited ecosystems (Kollet and Maxwell, 2008; Maxwell and Condon, 2016). Other studies stressed the connections between groundwater and the near surface climate through coupled numerical modelling experiments (Larsen et al., 2016; Wang et al., 2018). The shallow groundwater is also of importance in the urban context (Bricker et al., 2017), with special focus on urban flooding which can be directly induced or indirectly intensified by high groundwater levels (Jankowfsky et al., 2014; Kreibich

and Thieken, 2008; MacDonald et al., 2012). Moreover, MacDonald et al. (2010) and Upton and Jackson (2011) have studied the underlying processes, estimated return periods and mapped risk of groundwater flooding events.

In Denmark, the quantitative status of shallow groundwater systems is challenged by climate change and groundwater abstraction (Henriksen et al., 2008; Karlsson et al., 2016). In more detail, Kidmose et al. (2013) demonstrated that groundwater levels are expected to rise by up to 1.5 m for a 100 year event with respect to today's average conditions. Similar findings were

presented by van Roosmalen et al. (2007) who quantified regional differences across Denmark in the projected change of groundwater levels depending on soil types with more profound increases in high permeable sandy soils. Moreover, Henriksen et al. (2012) analysed climate change effects on the shallow water table over Denmark for mean and max conditions for nine different climate models and identified changes of at least 0.5 m for 26 % of Denmark. This finding represented the median change across the nine applied climate models.

The abovementioned problems call for comprehensive modelling tools that can support environmental decision making aiming at tackling today's and future's challenges related to the shallow groundwater. Spatial scales which are relevant for society and required for adequate decision making can typically not be provided by numerical, physically-based models alone. This limitation is mainly related to the fact that such models are computationally very expensive which hinders to conduct thorough calibration, sensitivity and uncertainty analysis at high resolution (Asher et al., 2015; Stisen et al., 2017). Further, there exists

a general difficulty to parameterize subsurface processes regardless the scale (Beven et al., 2015). Moreover, the wealth and detail of hydrological data is under continual growth (Chaney et al., 2018) and the resulting big data is often not harnessed optimally in existing modelling frameworks (Best et al., 2015; Nearing et al., 2016). As outlined by Reichstein et al. (2019), machine learning will play an essential role in advancing traditional modelling systems by integrating machine learning and numerical models. The development and testing of such hybrid models, complementing benefits of physically-based models



and machine learning, has gradually gained more attention in recent years in the hydrological community. A roadmap toward machine learning facilitated discoveries of hydrological systems has been outlined by Shen at al. (2018) and will likely play an eminent role in the future of Hydrology. More generally, Karpatne et al. (2017) coined the paradigm "theory-guided data science" which comprises a diverse list of approaches with which physically-based models and machine learning can be

combined. All three abovementioned references focus on the coupling of physically-based models with the versatility of data driven modelling frameworks. In more detail, they identified that the interpretability of machine learning models is among the main challenges for the successful adoption of big data technologies in the hydrological science.

This study highlights the applicability of machine learning, namely, the Random Forests (RF) algorithm (Breiman, 2001), to model the depth to the shallow groundwater at regional scale at high spatial resolution. The aim is to produce a map that

captures an extreme wintertime condition representing a minimum depth to the water table. Such an event can potentially induce groundwater flooding which poses risks related to infrastructure and agriculture. Thus, the resulting high resolution map will be a versatile screening resource for environmental decision making and climate change adaption planning. The proposed RF model draws on the Danish national water resources model (Højberg et al., 2013) which provides a coarse estimation of the depth to the shallow water table. In this way, the RF model utilizes the coarse prediction of the physically-

based model to ensure overall physical consistency, which may not be granted by the RF model alone. Hence, this study tests a simple hybrid mode integrating the output of a numerical model in a machine learning framework. Before machine learning techniques can build the toolbox of future's environmental decision making and planning, methods to conduct comprehensive sensitivity analysis and uncertainty assessment need to be developed and tested thoroughly. In order to advance this field of hydrological research, this study compares two different methods to quantifying uncertainty of a RF model, namely Random

Forest Regression Kriging (Hengl et al., 2015) and Quantile Regression Forests (Meinshausen, 2006). Furthermore, this study features a novel methodology to quantify covariate importance, and thereby the sensitivity of the model inputs, which ultimately helps to better comprehend and interpret the RF prediction.

Numerous studies have already successfully employed machine learning techniques to predict the temporal dynamics of the groundwater system based on artificial neural networks (Banerjee et al., 2009; Daliakopoulos et al., 2005; Shiri et al., 2013;

Yoon et al., 2011) or other techniques (Fallah-Mehdipour et al., 2013). Opposed, opportunities to utilize data driven modelling to assess the spatial dimension of the water table have so far not been fully exhausted. Fienen et al. (2013) are among the few that utilize a machine learning technique to map the depth to the groundwater in space (i.e. Bayesian network). Machine learning has already been applied to model other groundwater related variables such as, nitrate concentration (Nolan et al., 2015; Tesoriero et al., 2015), arsenic concentrations (Erickson et al., 2018; Winkel et al., 2011) or redox conditions in the

subsurface (Close et al., 2016; Koch et al., 2019), and the potential to map the depth to the water table is tangible.

The four main objectives of this paper are as follows: (1) to train a RF model that is capable to predict the depth to the shallow water table at high spatial resolution, (2) to outline a simple and generic method that unifies a physically-based model and machine learning, (3) to conduct a comprehensive sensitivity analysis to better interpret the RF model prediction, (4) to assess the uncertainty related to the RF model based on two different approaches.





## 2 Methods

### 2.1 Study Area

The study area encompasses a large part of the Danish peninsular, which is located in Northern Europe (54.5–57.8°N and 8.0–10.9°E) and referred to as Jutland. The extent of the study area amounts to approximately 15,000 km$^2$ and its general surficial
geological setting is illustrated in Figure 1. The landscape of Jutland was formed by the sequence of Pleistocene glaciations and postglacial processes. The geology of the eastern part is dominated by Weichselian moraine sediments with high clay content, whereas the west is characterized by moraine sediments originating from the Saalian age (Hill Island) intertwined by Weichselian sandy outwash plains.

### 2.2 Data

This study aims at modelling the depth to the shallow water table at 50 m spatial resolution using a machine learning modelling framework. Disregarding the prevailing temporal variability of groundwater dynamics close to the surface, we chose to model an extreme event that characterises a minimum depth, expected to arise every year. Based on the climate in Denmark, such an event normally occurs towards the end of winter, when shallow aquifer systems are replenished after several months of typically high rainfall and low evapotranspiration. Applying machine learning to model an extreme event of a highly dynamic
variable poses distinct challenges to the training dataset. Long timeseries of groundwater head measurements are scarce, and shallow groundwater time series is even more rare. In fact, many shallow wells, with screens within the uppermost 10 meters, provide just one to a few observations in total. In order to capitalize these low frequency sampled wells, we have developed a method to transform any given observations to an expected high water table. For this transformation, sinus curves were defined with varying amplitudes capturing the annual dynamics of the shallow groundwater for various hydrogeological settings. The
workflow is described in more detail below.



**Figure 1: The study site is located in central Denmark. The overview figure, in the uppermost right corner, depicts the digital elevation model. The main map shows the predominant geological landscape types. The training dataset contains observations at ~15,000 wells and ~15,000 additional data points placed along major rivers, lakes and coastline. The depth to the shallow groundwater is set to zero for the additional data.**

First, well data, covering the entire model domain, was extracted from the national database, Jupiter (Hansen and Pjetursson, 2011). Groundwater head observations from wells with a maximum filter depth of 10 m were assorted for a 20 year period between 1998 and 2017. Several constraints were applied to this initial extraction: (1) the mean water level may not be below filter depth, (2) the water levels may not exceed 5 m above surface, (3) the standard deviation of head observations may not be greater than 3 m and (4) the well may not be in operation. By applying these four constraints, 14,916 wells with one or more head observations were selected which approximately corresponds to a density of one well per km$^2$. Figure 1 shows the location of the wells.

Second, wells with more than five observations, of which 392 were present, were grouped according to their hydrogeological setting. Subsequently, their standard deviation was studied in more detail in order to define the sinus curve amplitudes for each of the groups. In total, 27 combinations, describing the general hydrogeological setting of a well, were assessed. These groups



were based on three categories with three sub-categories each, (1) permeability (high, low or unknown), (2) aquifer condition (confined, unconfined or unknown) and (3) proximity (near coast, near stream or other). The amplitude of the sinus curve was set to the 99% confidence interval and, under the assumption of normality, calculated as 2.576 times the standard deviation. Based on the analysis of the variability at 392 wells with long timeseries the average annual amplitude of the sinus curves

varied between 0.5 m and 1.5 m depending on the hydrogeological setting. The largest amplitude was associated to wells with filters in low permeable sediment, under unconfined conditions and not in the vicinity of coast or streams. Low amplitudes were generally connected to wells closer than 100m to streams, lakes or coastline. The minimum and maximum of the sinus curves was set to arise in the mid February and mid August, respectively.

Third, the groundwater head value describing an extreme wintertime condition at each well was defined twofold. At wells with

five or more observations the recorded minimum depth was used as input to the training dataset. Opposed, at wells with fewer observations, the predefined sinus model was applied to transform an observed minimum to an expected wintertime minimum. With this approach, the observed variability at wells with high quality data was used to infer a meaningful minimum value, describing an extreme wintertime situation, at wells with few observations. Figure 2 exemplifies this approach in more detail. The first two examples express wells with long timeseries, used to define the sinus amplitude corresponding to the well's

hydrogeological setting. Here, the minimum observations were extracted as input to the training dataset. The bottom two examples depict cases with few observations where a sinus curve was applied to transform the observed minimum depth to an expected winter minimum condition. In cases where the transformation resulted in negative values, i.e. manifesting artesian conditions, the value was set to zero. This correction was considered meaningful as many of these wells were located in unconfined conditions.

There are two sources of uncertainty that were not considered in this analysis. First, the observational uncertainty related to the head values in the well database was not considered. Second, the sinus model used to traverse any given observation to an expected wintertime minimum neglects seasonal and inter-annual variability.



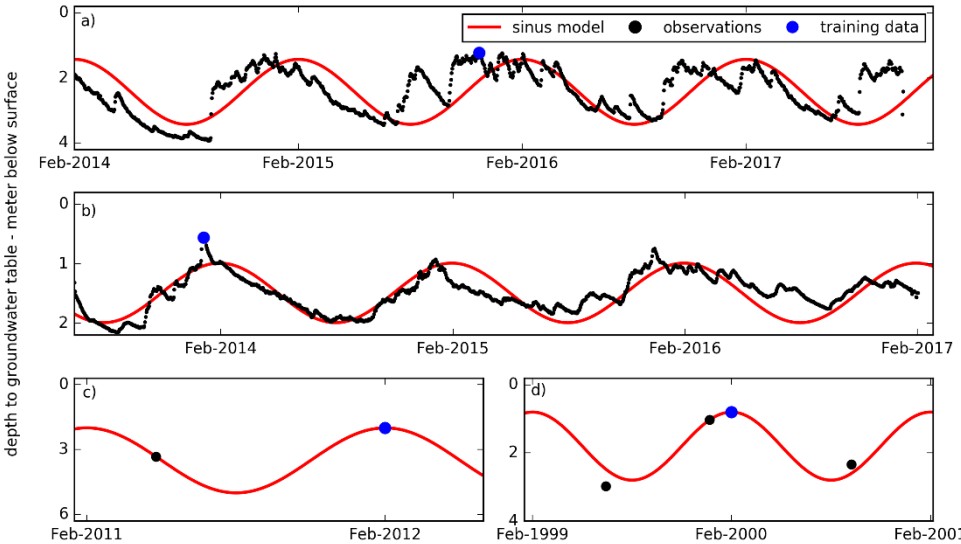

**Figure 2: Four examples showing how the training dataset was derived. At wells with more than four observations, a) and b), the minimum daily observation was chosen. Examples a) and b) represent long timeseries and sinus curves with amplitudes of 1 m and 0.5 m, respectively, that were used to describe the annual variability. Examples c) and d) represent two cases with few observations. Here, sinus curves with predefined amplitudes, 1.5 m and 1 m, respectively, corresponding to the well's hydrogeological setting were applied to traverse the observed minimum depth to an expected wintertime minimum.**

Additional observations were placed along streams, coastline and the centre points of lakes. At the given locations, the depth to the shallow groundwater was set to zero. This extension of the training dataset was found necessary in order to provide critical information to the machine learning model which was otherwise not conveyed in the borehole data alone. However, only a subset consisting of 1,900 random samples of the 16,210 additional observations was used for training of the machine learning model. This corresponds to approximately the same well density as found in the original training dataset, taking into consideration the combined area of streams, coastline and lakes in 50 m grid resolution. In this way, the information content of the well data and the additional data was balanced. The complete dataset of additional observations was however utilized in the uncertainty analysis.

In Table 1, a list of the environmental covariates used to model the depth of the shallow water table is found. In total, 26 covariates were assembled as input to the machine learning model. This list comprises information on soil texture, drainage conditions, geology, topography based characteristics, waterbody proximity, precipitation, land cover, geographic location and outputs from a hydrological simulation with the Danish national water resources model (DK-model: Højberg et al., 2013). The native spatial resolution of the covariates varied, but all covariates were resampled to 50 m to be in agreement with the defined output resolution. The covariates were subdivided into six groups, i.e. geology, topography, waterbody relation, precipitation,



land cover, coordinates and hydrological model. This subdivision was implemented in the sensitivity analysis of the machine learning model to eliminate correlations between covariates.

**Table 1: Overview of the covariates used to model the shallow water table using RF.**

| Variable | Source | Group |
|---|---|---|
| Clay content; 0-30 cm | Adhikari et al., 2013 | Geology |
| Clay content; 30-60 cm | | |
| Clay content; 60-100 cm | | |
| Clay content; 100-200 cm | | |
| Quaternary thickness | Binzer and Stockmarr, 1994 | |
| Depth to clay occurrence | Højberg et al., 2013 | |
| Drain probability | Møller et al., 2018 | |
| Drain class | Møller et al., 2017 | |
| Lowland classification | Aarhus University – Danish Centre for Environment and Energy | |
| Landscape typology | | |
| Georegion classification | Adhikari et al., 2014 | |
| Soil type | Geological Survey of Denmark and Greenland | |
| Digital elevation model | Danish Agency for Data Supply an Efficiency (SDFE) | Topography |
| Detrended digital elevation model | | |
| Topographic wetness index | Böhner and Selige, 2002 | |
| Saga wetness index | | |
| Flow accumulation | SDFE | |
| Slope | | |
| Vertical distance to nearest waterbody | | |
| Horizontal distance to nearest waterbody | | Waterbody relation |
| Waterbody (lake, river and coast) classification | | |
| Precipitation | Danish Meteorological Institute | Precipitation |
| Degree of urbanization | Danish climate change adaption portal | Land cover |
| Land cover | CORINE Copernicus | |
| Coordinates (utmx) | N/A | Coordinates |
| Coordinates (utmy) | | |
| DK-model; depth to max groundwater level | Henriksen et al. 2014(Højberg et al., 2013)(Højberg et al., 2013)(Højberg et al., 2013)(Højberg et al., 2013)(Højberg et al., 2013)(Højberg et al., 2013)(Højberg et al., 2013) | Hydro model |

## 2.3 Random Forests

5    This study applied Random Forests (RF) regression to model the depth to the shallow water table at high spatial resolution at regional scale. RF was first proposed by Breiman (2001) and emerged to a prevalent modelling tool covering a wide range of geophysical and environmental contexts. These include, among others, digital soil mapping (Hengl et al., 2015; Ließ et al.,



2012), estimating nitrate pollution in aquifers (Rodriguez-Galiano et al., 2014; Tesoriero et al., 2017), biomass estimation using satellite remote sensing (Mutanga et al., 2012), landslide susceptibility analysis (Youssef et al., 2016), mineral prospectivity mapping (Rodriguez-Galiano et al., 2015) or estimation of young water fractions across catchments (Lutz et al., 2018). RF has proven to provide high predictability for multivariate modelling of complex, non-linear variables and multiple

benchmarking studies have documented the capabilities of RF to outperform other machine learning techniques (Nussbaum et al., 2018; Rodriguez-Galiano et al., 2015; Youssef et al., 2016). Like other data driven modelling approaches, training is an essential step in the RF model building process. Based on the training dataset, RF learns linkages between the covariates and the target variable at sampled locations, which then are generalized to make predictions at unsampled locations. The core of RF is an enhanced utilization of decision trees. More precisely, RF builds an ensemble of decision trees, where each tree

recursively splits the training data into more homogenous groups. The formulation of the decision trees contains two elements of randomness with the aim to increase the diversity within the ensemble of decision trees. First, the concept of bagging is applied. Bagging is an ensemble technique, which generates a unique bootstrap sample of the original training dataset for each decision tree. Based on sampling with replacement, each bootstrap sample contains approximately 66 % of the original training data. The average across the entire ensemble represents the final RF prediction. Second, only a subset of covariates is drawn

upon when splitting the data during the process of decision tree building. This subset, usually 33 % of the available covariates, is selected randomly for each split. In combination, the two elements of randomness decrease the accuracy of a single tree; however the diversity between the trees increases which results in a robust prediction when averaging across all trees.

The bootstrapping procedure divides the training data into an in-bag part, which is used for building the decision trees, and an

out-of-bag (oob) part, which is excluded from the training. This partitioning is unique for each tree of the ensemble and thereby provides a valuable internal cross validation test. In other terms, each tree can be validated with a different oob sample and the average across all oob predictions allows to quantify the overall accuracy of the RF model. We applied the python package Scikit-learn (Pedregosa et al., 2012) to conduct the RF modelling for this study.

## 2.4 Sensitivity Analysis

The concept of covariate permutation allows to assess the importance of each covariate, acting as input to a RF model (Biau and Scornet, 2016). This can be understood as a sensitivity analysis which can help to better comprehend and interpret the trained RF model and to gain physical insights into the otherwise intransparent black box model. This is achieved by permuting each covariate at a time, while leaving the remaining covariates unchanged, and tracing the apparent decrease in oob validation metric, typically the coefficient of determination. This concept is common practice to assess covariate importance for a trained

RF model (Ließ et al., 2012; Lutz et al., 2018). However, this analysis is limited to the training dataset and conclusions on which covariates dominate the prediction and how this varies spatially cannot be drawn. In order to gain insights into the spatial patterns of covariates importance, we have developed a novel method, which applies the abovementioned concept of covariate permutation on the prediction dataset instead of the training datset. The aim of the sensitivity analysis is to identify




a relative ranking of covariate importance for each simulation grid, which can ultimately provide an increased interpretability for the users. The starting point of the analysis is the trained RF model and its prediction for all simulation grids. Sequentially, each covariate is permuted, while leaving the remaining covariates unchanged, and the trained RF model is used to make a modified prediction. The difference between the modified and original prediction is recorded. The cycle of permutation and

prediction is repeated *n* times until the mean difference across *n* permutations converges for each simulation grid. This is necessary, because a single permutation may allegedly result in no or minor change in covariate value at specific grids. Once the mean difference has converged, the covariates can be ranked with respect to their associated mean difference for each simulation grid. This ranking expresses the relative covariate importance and is the key result of the proposed sensitivity analysis. Maps showing the top ranks can be used to visualize the spatial patterns of the sensitivity of the RF model.

Typically, strong correlations are found between covariates, which may result in an alleged low importance when being permuted individually (Koch et al., 2019). In order to overcome this limitation we suggest a supplementary analysis that collectively permutes groups of covariates that are physically related.

**2.5 Random Forests Regression Kriging**

Extending RF with geostatistical methods is gaining popularity in the field of digital soil mapping (Guo et al., 2015; Hengl et

al., 2015) and related environmental modelling studies (Ahmed et al., 2017; Li et al., 2011; Viscarra Rossel et al., 2014). Regression Kriging (RK) is a widely applied approach that combines a multiple linear regression (MLR) model with a geostatistical model of the MLR residuals (Hengl et al., 2007; Odeh et al., 1995). In order to integrate RF into RK, RF can simply replace the MLR model. In this way, RF provides an overall data-driven trend and the RF residuals can be interpolated using geostatistics. This results in a hybrid model that is commonly referred to as Random Forests Regression Kriging (RFRK).

To our knowledge, RFRK has not yet been applied with the purpose to predict a hydrological state variable such as groundwater head. RFRK can be expressed by

$$P_{RFRK}(s_0) = T_{RF}(s_0) + \hat{e}_{RF}(s_0), \quad \text{(eq.1)}$$

where $T_{RF}(s_0)$ is the RF prediction at location $s_0$ and $\hat{e}_{RF}(s_0)$ is the estimated residual at the same location. The sum of trend ($T_{RF}$) and residual ($\hat{e}_{RF}$) yields the final RFRK prediction ($P_{RFRK}$). This study utilizes kriging to interpolate the oob residuals

of the RF model. Kriging is a popular geostatistical technique for spatial interpolation that employs knowledge about the spatial autocorrelation of a variable, which can be captured by a variogram model. For the definition of a variogram model, the omnidirectional empirical semivariance ($\gamma$) is calculated by

$$\gamma(h) = \frac{1}{2n(h)} \sum_{i=1}^{n(h)} [e(s_i) - e(s_i + h)]^2, \quad \text{(eq.2)}$$

where $n(h)$ marks the total number of data pairs at a given lag distance $h$. $e(s_i)$ represents the oob residual at location $s_i$ and

$e(s_i+h)$ is the residual separated by lag $h$ from $s_i$ (Matheron, 1963). A variogram model is fitted to $\gamma$ to model the spatial autocorrelation structure of the oob residuals (Deutsch and Journel, 1998). The parameters defining a variogram model are



type, range, sill and nugget. The R package Gstat (Pebesma, 2004) was applied for variogram modelling and kriging interpolation.

The addition of residual kriging to RF results in high accuracy at grids coinciding with observations. Furthermore, kriging quantifies the prediction uncertainty following the defined variogram model. Generally, the kriging variance is low in vicinity to data points and increases to the sill value once the distance to the nearest data point is beyond the range of the variogram model.

### 2.6 Quantile Regression Forests

Using traditional RF, the prediction is obtained by averaging across the ensemble of decisions trees. This disregards the distribution of the target variable originating from several hundreds to thousands of decision trees, which are typically necessary to build a robust RF model. Meinshausen (2006) developed the Quantile Regression Forests (QRF) method that analyses the quantiles of the distribution of the target variable at prediction grids. This results in an estimation of prediction uncertainty or prediction intervals. The latter is obtained by recording specific quantiles which mark the lower and upper confidence limits (Hengl et al., 2018). The adoption of QRF for hydrological variables is still gradually and only few studies documented its applicability (Francke et al., 2008; Zimmermann et al., 2014). To our knowledge QRF has not been applied to quantify uncertainty of groundwater level predictions.

### 2.7 Error Propagation

The use of error propagation allows combining several sources of uncertainty. In order to apply this concept, no significant covariance between the uncertainties may be present. Following this assumption, the uncertainties ($\sigma$) of different sources (*x, y,* etc.) can be combined by the following approach:

$$\sigma_{Combined} = \sqrt{\sigma_x{}^2 + \sigma_y{}^2 + \cdots} .$$ (eq.3)

The key limitation is that the underlying assumption seldom holds (Refsgaard et al., 2007). Nevertheless, this propagation approach can facilitate a simple and powerful screening analysis (Kidmose et al., 2013).

### 3 Results

### 3.1 Random Forests Model

For the purpose of modelling the depth to the shallow water table at 50 m spatial resolution for an extreme wintertime minimum event, a RF model was trained using the 26 available covariates and groundwater head data. The training data comprised ~15,000 wells and 1,900 additional observations placed along streams, coastlines and in lakes. After initial testing, the RF model was parametrized as follows; the number of decision trees was set to 1,000, bootstrapping with replacement was applied to sample the training data, 33% of the covariates were considered to identify the optimal data split, trees were fully expanded





and thus not pruned, the mean squared error was selected as criterion to identify the optimal data split and regression was chosen as method.

Figure 3 depicts the internal cross-validation test based on the oob samples. The oob prediction can be considered as an independent validation test and three performance metrics, i.e. coefficient of determination ($R^2$), mean absolute error (MAE)

and root mean squared error (RMSE) indicated an overall good performance. More than half of the variance contained in the training data was captured by the RF model, the MAE amounted to 79 cm and the RMSE was 1.19 m. The density scatter plot in Figure 3 zooms into the top 6 m and it becomes apparent that very shallow observations (< 0.5 m) were systematically biased while deeper observations were estimated in good agreement, closely to the 1:1 line.

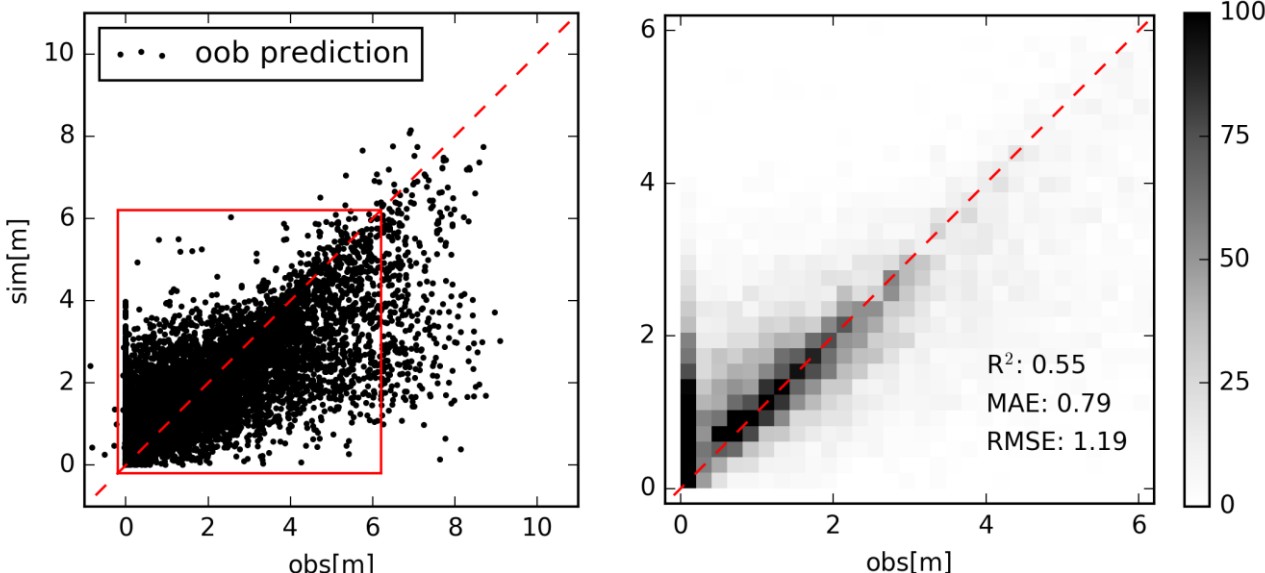

**Figure 3: The RF validation test was performed based on the out of bag sample technique. The axes depict the simulated and observed depth to the shallow water table. The left panel displays a standard scatter plot containing ~17,000 data points. The right panel shows a zoom (extent indicators in red in left panel) and data is visualized as a density scatter plot. The colourbar represents the number of data points in each square.**

Post training, the RF model was utilized to predict the depth to the shallow water table and the resulting map is shown in

Figure 4. Regional patterns of the shallow water table were estimated as expected with deeper water tables in parts of the sandy meltwater plains in the western part of the domain and a water table that was overall close to the surface in the moraine landscape, as shown in Figure 1. Areas with low topography were generally exposed to a very shallow water table, which also corresponded to the conceptual understanding of the system. The 50 m spatial resolution provided a very detailed picture of the spatial patterns associated to the water table and the complex interplay of the covariates became apparent. This is shown

on the basis of two zoom extents, highlighting urban areas, in Figure 4. The stream network and lakes are clearly visual with a depth of zero, which indicates that appending the additional observations to the training data resulted in the intended affect. The severity of the risk of groundwater induced flood events becomes apparent through the statistics of the RF map. The mean



depth to the groundwater for a typical wintertime minimum event constituted 1.9 m for the entire modelling domain. Around 29 % of the domain was characterised by a depth to the shallow groundwater lower than 1 m and a depth of 50 cm or less was evident for 14 % of the area.

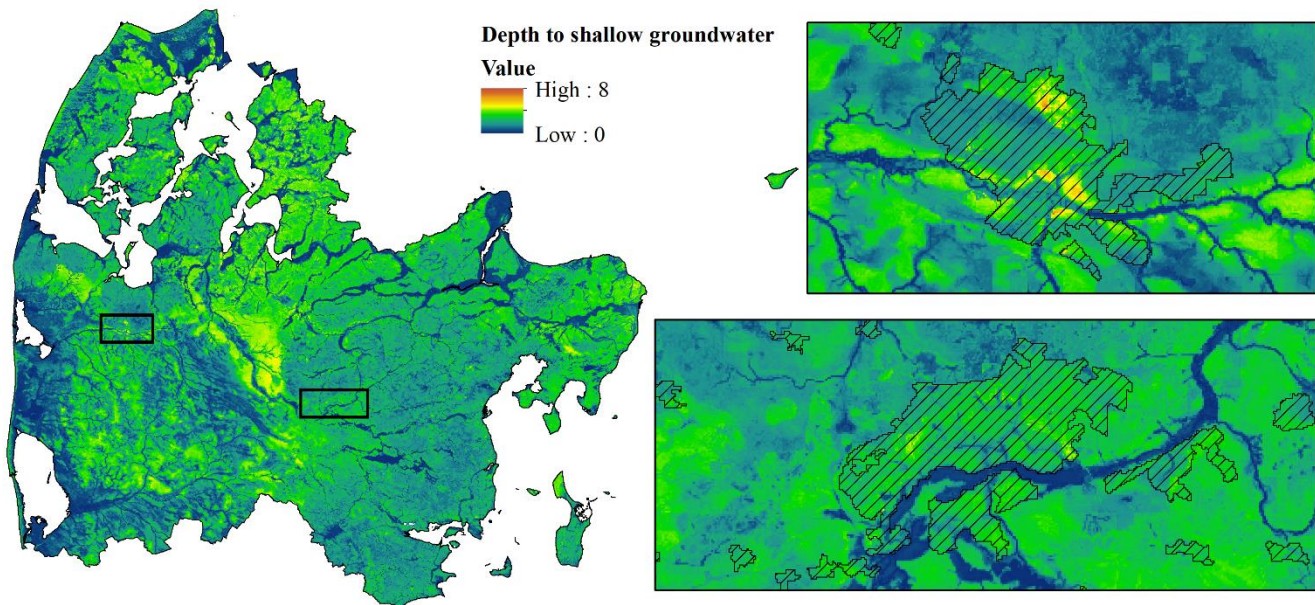

**Figure 4: The resulting map of the depth to the shallow water table in 50 m grid resolution. The zoom extents highlight two urban areas. The top zoom displays the city of Holstebro and the bottom zoom depicts the city of Silkeborg. Urban areas are visualized in hatch signature.**

### 3.2 Sensitivity Analysis

Sensitivity analysis, i.e. assessment of covariate importance, was performed for the trained RF model and for the RF prediction. Figure 5 shows the results for the former. The vertical distance to the nearest waterbody was the dominant covariate for the simulation of the shallow water table. A decrease of 60 % in performance was apparent when the variable was permuted. We found a direct relationship between the two variables, which highlighted that the shallow groundwater did not explicitly follow terrain variability. This resulted in a relatively deep water table at locations where the vertical distance is high and vice versa. The second most important variable in the trained RF model was the simulated water table by the national water resources model (DK-model), associated with a 15 % drop in performance when being permuted. The DK-model provides a typical minimum depth to the shallow water table at 500 m resolution for a 20 year reference period (1991 – 2010). This indicated that the DK-model could supply a valuable coarse trend to the RF model.

Figure 5 also quantifies the importance of physically related covariates. When permuted collectively, covariates associated to the topography resulted in a decrease of nearly 100 % in performance and thereby, the respective covariates formed the most important group. They were followed by covariates describing the waterbody relation (~70 % drop in performance) and



geology related variables (60 %). As the vertical distance to the nearest waterbody relates to both, topography and waterbody proximity, it was included in both groups.

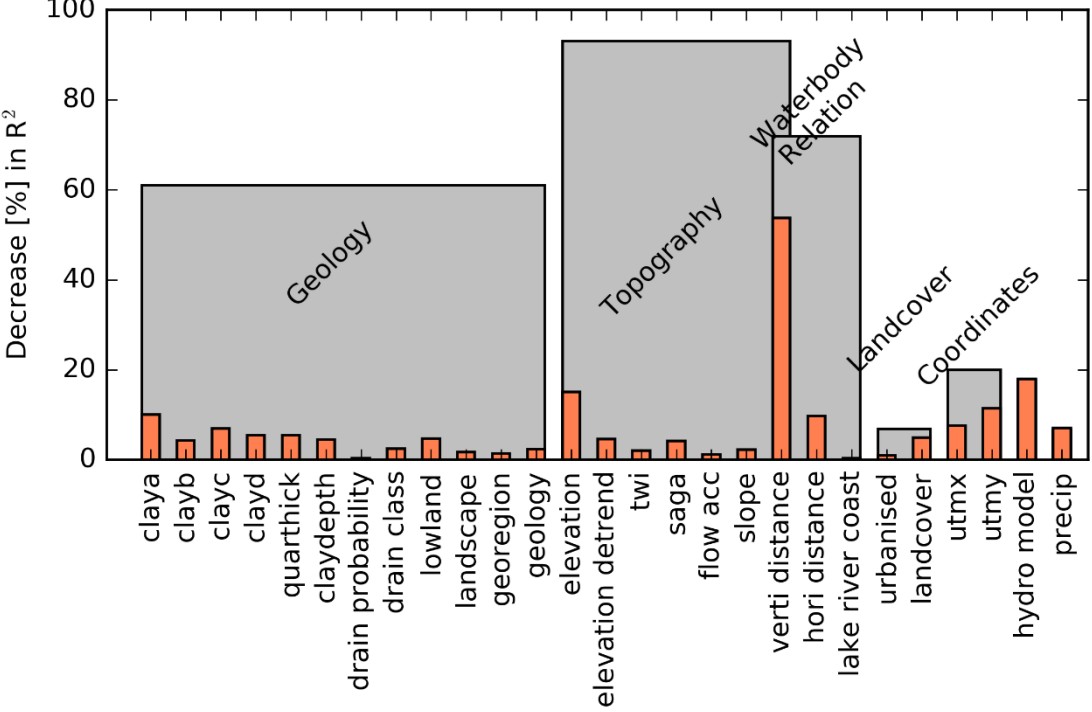

**Figure 5. Variable importance of the trained RF model. The concept of permutation accuracy was implemented to quantify the decrease in out-of-bag performance R$^2$. Permutation was applied not only to single covariates (orange) but also to groups of covariates (grey).**

Figure 6 depicts maps of the top two most important covariates for the RF prediction. Covariates were permuted collectively following the groups presented in Table 1 and as applied in the sensitivity analysis of the trained RF model (Figure 5). Each covariate group was permuted 250 times to ensure that the difference to the original RF prediction converged at the individual grids. The simulated water table in the moraine landscape in the eastern part of the model domain was controlled by covariates related to the geology. Here topography is gently undulating and sediments are clay rich, which, in combination, resulted in a water table close to the surface with small-scale variability caused by geological heterogeneity. The second most important covariates in the moraine landscape were mainly the DK-model or the UTM coordinates. This underlined the complexity of the shallow water table in this landscape. The DK-model includes a comprehensive analysis of the entire system, taking the interplay between several factors, hydrogeology, topography, climate and others into consideration. In the RF model, coordinates provided the only possibility to assign uniqueness to a simulation grid, which was required in the moraine landscape to capture the complexity of the shallow water table. Topography was important at locations close to sea level or areas that were generally plane. Waterbody relations played an important role at location that were either very far away or very



close to a waterbody. Data on the location of urban areas, which was contained in the land cover group, was rated important for urban areas with moraine soils. In such clayey conditions, the subsurface is often drained resulting in a deeper water table. Overall, the importance of the DK-model appeared to be very local and generally scattered across the domain, which underlined the relevance of this covariate, as it could provide coarse information at locations where the standard covariates fail at providing

5    a meaningful generalisation.

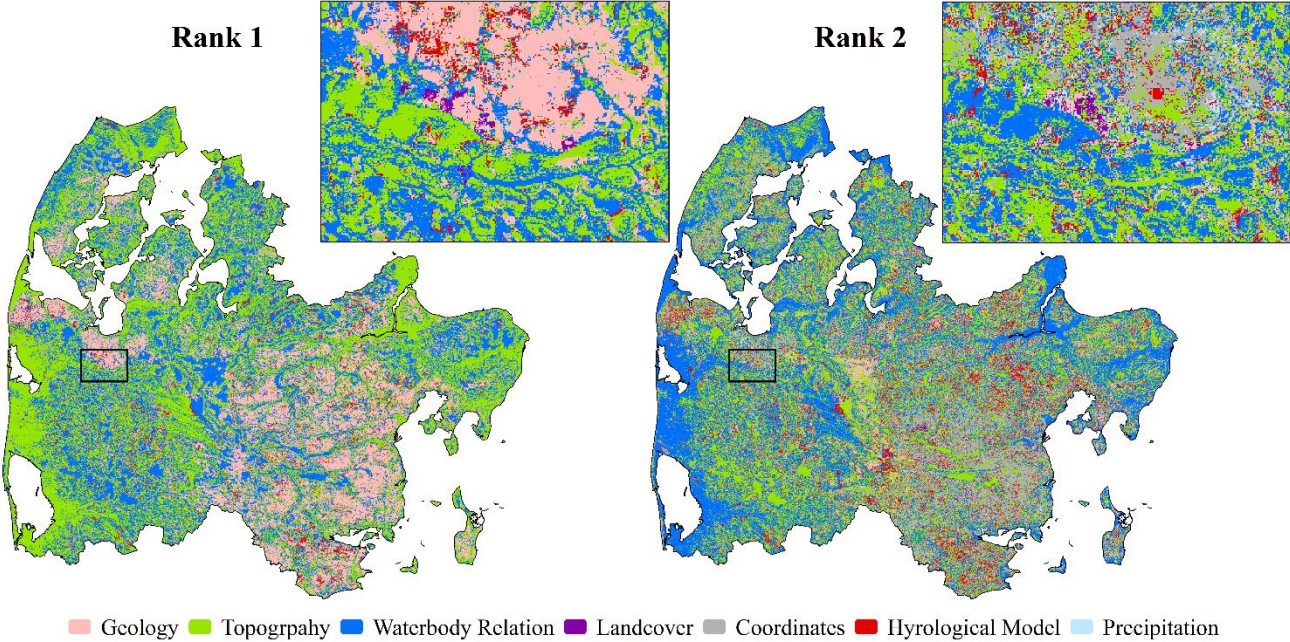

**Figure 6: The results of the sensitivity analysis are shown for the most sensitive covariate group (Rank 1, left panel) and the second most important covariate group (Rank 2, right panel). The city of Holstebro is chosen as zoom for both maps.**

### 3.3 Uncertainty Analysis

10   For the uncertainty analysis, we employed two methods, namely RFRK and QRF. For the first, the RF residuals were interpolated using kriging. Figure 7 shows the variogram model which was used in the kriging interpolation. The nugget was set to 0.26 $m^2$ and sill was defined as 1.02 $m^2$. An exponential variogram with a range of 700 m gave the most satisfying fit to the experimental semivariances calculated at 200 m lag distance.





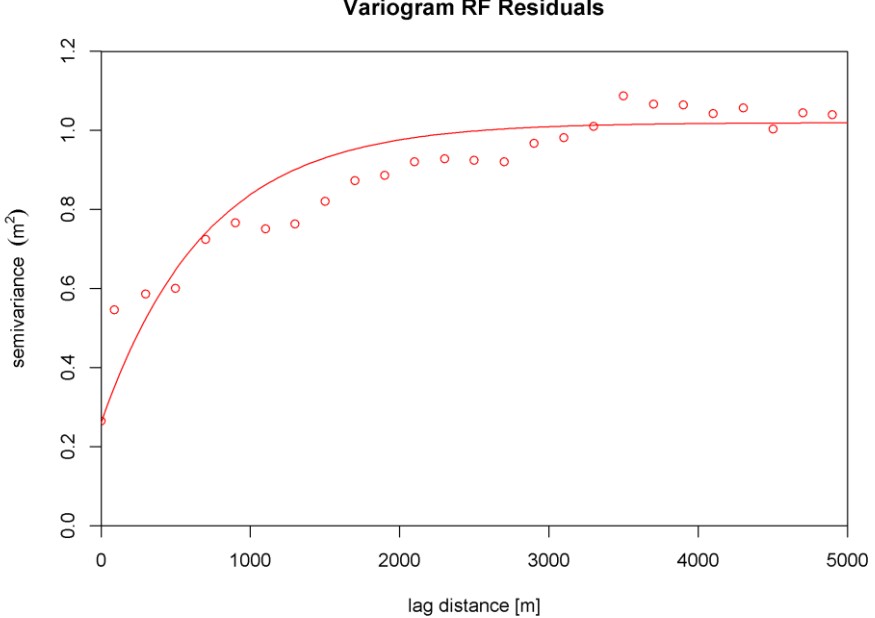

**Figure 7: The computed semivariances for the RF residuals (circles), based on the oob prediction. The line expresses the fitted variogram model.**

Figure 8 depicts the resulting uncertainty, which was expressed by the standard deviation for both of the applied methods. The

5 RFRK employed all available data, ~15,000 wells and ~15,000 additional observations along streams, coastlines and in lakes

in the RF residual interpolation. Following the predefined variogram, uncertainty was low in vicinity of an observation, which

increased with distance until the sill value was reached. Based on QRF, the derived uncertainty shows a different picture. Here

the uncertainty was expressed as the standard deviation of the 1000 individual decision tree predictions at each simulation grid.

In general the uncertainty estimated by RFRK was lower than QRF. In the western part, high uncertainties were in general

10 associated to locations with a large depth to the shallow groundwater and vice versa for the QRF based assessment. However,

the moraine landscape in the eastern part was characterised with an overall high uncertainty despite having an overall water

table that is close to the surface. Such physical dependencies that relate to the structure of the RF model were not captured by

the RFRK approach, which purely reflected borehole proximity. Therefore, we argue that the two approaches, to assess the

uncertainty of the RF model, can be considered independent. Based on the concept of error propagation (eq. 3), we could take

15 both sources of uncertainty, namely observation proximity and RF model structure, into consideration. Figure 8 presents the

map depicting the combined uncertainty. The mean uncertainty across the domain amounted to 0.92 m for RFRK, 1.68 m for

QRF and 1.93 for the combined map.




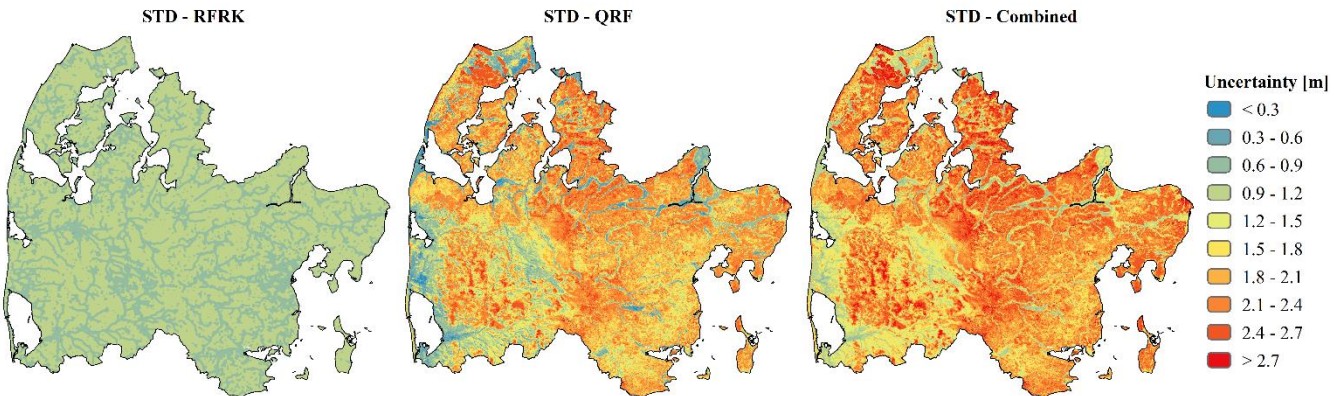

**Figure 8: Two methods to quantify uncertainty of a RF model are implemented: RFRK (left panel) and QRF (middle panel). Under assumption of independence, they can be combined and results are depicted in the right panel. For all maps, uncertainty is expressed as the standard deviation (STD).**

## 4 Discussion

### 4.1 Training Dataset

In order to capitalize undersampled wells, this study utilized sinus curves, with amplitudes fitted to observations at wells with long timeseries according to their hydrogeological setting. Even though this step introduces uncertainties, it was essential to generate a training dataset large enough to make robust predictions. Applying the same amplitude every year does not distinguish between dry and wet years, which is a clear limitation of the approach. The sinus curves describe an average seasonal variation within a hydrogeological class of boreholes and are thus not designed to reflect the variability of all boreholes within each class. Nevertheless, it is critical that the dataset used to train a RF model contains a wide range of observations before the model is able to generalize and make predictions. Along these lines, a training dataset can be expanded based on expert domain knowledge to capture otherwise underrepresented conditions (Koch et al., 2019). In this study, additional observations along streams, coastline and in lakes were appended to the training dataset with a depth to the water table of zero. In regions where the connection between surface and groundwater is generally good, like it is for this case in Denmark, the extent of surface waters can be considered a reliable proxy of the shallow water table. The additional observations used in this study guided the RF model to produce more reliable predictions.

The RF model was trained to a single event and thereby disregarding the temporal dynamics of the shallow groundwater system. Being designed as a simple screening tool, this can be considered an advantage; however, much of the complexity is not considered which is clear a shortcoming of the proposed method. In future work, the sinus model can be replaced by the Danish national water resources model (DK-model) to correct the undersampled wells more physically-based. Furthermore, this would allow a stringent extreme value analysis of the water table as well as climate change analysis. This was not included in this project as the resolution of the current DK-model is 500 m, which is too coarse to clearly differentiate temporal dynamics



between hydrogeological settings at a scale reflecting the location of a well. In the coming years, the DK-model will be updated based on recent hydrogeological interpretations and reconstructed in 100 m spatial resolution. This is expected to improve the predictability of the shallow water table and should then be utilized to update the RF model.

## 4.2 Random Forests Model

This study utilized the oob prediction to validate the performance of the RF model based on three metrics, namely coefficient of determination ($R^2$), mean absolute error (MAE) and root mean squared error (RMSE). The metric scores were overall very satisfying and in the range of what could be considered very acceptable in traditional groundwater flow modelling (Henriksen et al., 2003). These findings underpin the applicability of RF to model complex, non-linear variables with a validity that is difficult to obtain with traditional physically-based models. On the other hand, the validation test identified a systematic bias

of the trained RF model that was affecting wells with groundwater levels close to the terrain. The biased wells were predominately placed in clayey moraine sediments, which indicated location specific shortcomings of the RF model. The geology of the moraine landscape is heterogeneous which impacts the hydrogeological setting and thereby also the shallow groundwater (Xiulan He et al., 2014, Xin He et al., 2015). Moreover, some of abovementioned wells are placed in confined conditions, which in combination with the heterogeneous geology may hinder good performance of the RF model.

Studying the covariate importance identified the water table simulated with the DK-model at 500 m resolution as the second most important RF input. These results were very promising as the applied RF framework forms a straightforward implementation of unifying machine learning and physically-based models. More precisely, RF buildt upon the coarse DK-model using high-resolution covariate information which ensurined physical consistency.

Some covariates, e.g. drainage characteristics, topographic wetness index, were assigned an unanticipated low importance in
the sensitivity analysis of the RF model (Figure 5). This may indicate covariate redundancy or the fact that the metric to quantify covariate importance, decrease in the coefficient of determination, is not very sensitive to the permutations which may result in changes of wells with a very shallow water table. Future work must address the issue related to the choice of metric more systematically.

The resulting spatial resolution of 50 m provides a valuable screening tool for water management purposes. The risk of
groundwater floods on agricultural fields or urban areas is typically very local and driven by small-scale variations of topography and geology. This makes high-resolution predictions inevitable in order to reliably tackle related challenges. At regional scale, the 50 m resolution would not be feasible with traditional numerical modelling, which emphasizes the versatile applicability of RF. Many covariates are available at finer resolution and, as computational power becomes more and more dispensable, RF predictions at even higher resolution are within reach. This development should also build upon current
improvements of physically-based models, which are now already capable of providing results at resolutions in the range of hundred meters (Ko et al., 2019; Wood et al., 2011) and, thereby, such models could provide valuable trends, used as covariate in machine learning models.



This study proposed a novel approach to quantify covariate sensitivity of the simulation dataset, which results in a relative ranking of the most important covariates at grid level. This analysis provided physical insights on the driving mechanisms and, in general, the findings corresponded to the conceptual understanding of the hydrogeology in that region. Such sensitivity maps are extremely valuable for both, the modeller and the stakeholders working with RF predictions. The former group can

validate the physical consistency of the otherwise non-transparent black-box model and the latter will have a better understanding and ultimately also a greater acceptance of the predictions. However, care must be taken, because the permutations must convergence in order to provide a robust sensitivity assessment.

### 4.3 Uncertainty assessment

This study assessed the capabilities of RFRK and QRF to estimate uncertainties associated to a RF model that predicts

groundwater levels. Uncertainty was expressed by the standard deviation; alternatively, both methods could also be utilized to map upper and lower uncertainty bounds that represent certain confidence intervals. The key differences between the two proposed methods were as follows: (1) the uncertainty estimation of RFRK was in general lower than QRF and (2) the spatial patterns were diverging; RFRK reflected solely borehole proximity whereas QRF manifested a physical dependency of the uncertainty estimation. These findings are in line with recent comparison studies focusing on QRF and RFRK from the digital

soil mapping literature (Szatmári and Pásztor, 2018; Vaysse and Lagacherie, 2017). Szatmári and Pásztor (2018) argue that RFRK based uncertainty estimations are limited because results do not depend on the data value and therefore, the method expresses an unconditional variance. This stringent assumption of homoscedasticity, i.e. constant error variance, could be unrealistic for variables where the variance behaves proportionally to the measured value (Hengl et al., 2018). Moreover, RFRK assumes that the RF prediction, which is used as trend, is certain and thus, the kriging variance only reflects the distance

to the nearest observation. This assumption is too optimistic, as the uncertainty in the RF prediction is neglected. Once the training dataset is processed, RF disregards any uncertainties associated to the values of the target variable. In this study, uncertainties could originate from the applied sinus model used to transfer the observations to a typical wintertime minimum depth as well as the observations itself. In contrast, a physically-based hydrological model allows more transparency, as biased observations will be marked as outliers in the validation step. However, a data driven model, as flexible as RF, will incorporate

such outliers and thus biased predictions may arise.

As stated by Vaysse and Lagacherie (2017), QRF quantifies information of where a simulation point is located in the covariate space. In this way, QRF properly discriminates groundwater conditions of contrasted physical complexities of which some are better constrained by the training dataset than others. We argue that the RFRK shortcomings of assuming certainty in the trend prediction can be alleviated by the addition of QRF, which can capture the uncertainty of the RF model structure. Following

this hypothesis and assuming independence, error propagation can be applied (eq. 4). The combined uncertainty reflects both, uncertainty due to spatial proximity to the nearest observation, as provided by RFRK, and uncertainty induced by model structure, as quantified by QRF. In summary, RFRK captures uncertainty related to the geographical space whereas QRF describes uncertainties related to the covariate space.





Reducing uncertainties can be achieved by collecting more observations and thus expanding the training dataset. Especially in the eastern part of the domain, which is characterized by a high clay content and a heterogeneous surficial geology, additional data likely reduce the uncertainty. A measuring campaign in wintertime, when the shallow groundwater system is fully replenished, would be very beneficial to advancing the modelling capabilities. Additionally, a higher spatial resolution may

contribute to an uncertainty reduction, as observations can be represented more uniquely by the covariates.

In more general terms, as the numbers of hydrological applications based on machine learning are vastly expanding, standards on how to conduct uncertainty analysis must be formalized in the same way this has been done for numerical modelling (Refsgaard et al., 2007). Ultimately, such a development conditions the stakeholder acceptance of machine learning results.

## 5 Conclusions

This study focused on using RF to predict a map that depicts the depth to the shallow groundwater at 50 m resolution for a typical wintertime minimum. More precisely, a minimum event that is expected to occur annually and poses risk of groundwater flooding affecting both, urban areas and agricultural fields. The regional map will be extremely valuable for water resources management. We draw the following main conclusions from our work:

1.  RF is a versatile modelling tool with high accuracy that allows to model spatial detail beyond the possibilities of
traditional, physically-based, numerical modelling. The depth to the shallow water table was modelled with a mean absolute error of 79 cm for an independent validation test.

2.  Predictions from a coarse physically-based model that represent an overall trend of the water table can be utilized by RF as covariate. In this way, RF ensures physical consistency at coarse scale and exhausts high resolution information from topography, geology and other relevant variables. The DK-model at 500 m resolution was rated the second most
important covariate in the trained RF model, indicating that this simple form of unifying machine learning and physically-based modelling has great potential.

3.  The novel approach to assess covariate sensitivity for the prediction dataset goes beyond the standard applications where covariate importance is solely quantified for the training dataset. Results provide valuable insights on the spatial pattern of covariate sensitivity and can contribute to generate acceptability among end-users. The increased
interpretability of the RF predictions can reassure modellers by comparing the derived sensitivity patterns with their conceptual understanding of the system.

4.  In the general context of hydrological machine learning applications, more experience must be gained on how to properly quantify uncertainty. RFRK was found useful to assess observational proximity, but assuming certainty in the RF predications was regarded a shortcoming. This can be compensated by QRF, which is capable of addressing
the uncertainty related to the structure of the RF model. Simple, uncertainty propagation can be utilized to combine both methods under the assumption of independence. However, methods to take the uncertainties related to the observations itself and possible pre-processing of the training dataset are still lacking.





**Acknowledgments**

The work has been carried out with financial support granted by the Coast to Coast Climate Challenge project funded by the EU's LIFE programme.

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
