# Peer review of "Modelling of the shallow water table at high spatial resolution using Random Forests."

_Hydrology and Earth System Sciences, 2019_

## Referee Comment (RC1) · Anders Bjørn Møller (Referee) · 11 Jun 2019

General comments

I have read the manuscript "Modelling of the shallow water table at high spatial resolution using Random Forests" submitted to HESS by Koch et al., in order to provide a referee comment.

The manuscript is well structured, clear, concise and well written. It addresses the depth to the shallow water table, which is a highly relevant issue, and introduces a number of novel methods in doing so. Some parts of the introduced methods have great potential, not only for hydrological applications but for spatial predictions with machine learning in general.

[Figure]

My main concerns with the manuscript lie with some of the specific choices that the authors make in implementing the methods, especially related to the assessment of the accuracy and uncertainties of the predictions. I will elaborate on these concerns in the following section. However, given that the authors address them, the manuscript is highly suitable for publication in HESS.

Specific comments Firstly, I am wondering why the authors choose to map the depth to the shallow water table rather than the elevation of the shallow water table. I would expect the elevation of the shallow water table to show less spatial variation than the depth from the surface. It should therefore be easier to predict, all other things equal. I am sure the authors have good reasons for this choice, but I would like to see them stated explicitly.

Secondly, I would like to comment on the use of a sine function to model an annual minimum event. I think it is a useful and generally robust way to address the issue of working with limited data. However, the method could be improved upon in a number of ways. Firstly, the maximum of the curve does not match the maximal observed water levels. The authors could therefore have calculated the uncertainty related to the sine model and, ideally, used these uncertainties in the Random Forest model. The authors already state this in the manuscript, but my second comment is related to the same issue. For training locations with sparse data, the authors used the maximum of the sine curve, but for training locations with more observations, the authors used observed maximum water levels. This choice muddles the results, both in terms of the predicted values and their accuracies. Is it a map of the expected minimum depth to the shallow water table, averaged over a number of years? Or is it a map of an extreme event, observed only in some years? The mixture of training data makes this question difficult to answer.

Thirdly, I have concerns about the way that the authors assess the accuracy of the predictions. The training dataset shows a high degree of clustering. Therefore, when the authors use the out-of-bag predictions for assessing the accuracy, the points used for

assessing the accuracy will be located close to the training points used for making the predictions. It is very likely that the values are spatially autocorrelated, and the stated accuracy is therefore probably not representative for the study area as a whole. I would expect the accuracy to be lower for the parts of the study area that do not have a high density of observations. A spatially structured accuracy assessment, as proposed for example by Muscarella et al. (2014), would most likely provide a more representative accuracy assessment. Furthermore, I am wondering if the authors used all the training points for the predictions. The training dataset contained both groundwater and surface water observations. However, the aim is not to predict surface water levels, and I would therefore say that one could justify removing the surface water points from the out-of-bag predictions when assessing the accuracy.

Fourthly, I very much like the way that the authors handle covariate importance. Being able to assess covariate importance in geographic space is potentially extremely useful, for both researchers and end users. However, I do not think that decrease in R2 is the best measure of covariate importance. One can potentially obtain a high R2, even if the absolute values are inaccurate. A better choice would therefore be to assess the relative change in a measure that accounts for absolute values, such as RMSE, Lin's concordance or the Nash-Sutcliffe efficiency.

Fifthly, while I appreciate that the authors assessed the uncertainties of the predictions in two different ways, I do not think that combining them is justified. The theoretical basis for the approach seems scarce. Both the RF uncertainties and the residuals used for kriging relate to the same model, and it is therefore a stretch to say that they are independent. Furthermore, quantile regression forest should be able to assess uncertainties quite accurately without any further elaboration, as shown for example by Rudiyanto et al. (2018). I think a large part of the spatial autocorrelation in the residuals would disappear, if one takes into account the uncertainties related to the RF predictions. The uncertainties in the predictions make the residuals uncertain as well, which complicates regression-kriging. When experimenting with techniques, as

the authors do, it is important to set aside an independent part of the dataset to be able to assess the accuracy of the estimated uncertainties. However, the authors do not do this, and it is therefore impossible to assess if the error propagation actually leads to a better estimate of the uncertainties. Unless the authors can adequately adress these shortcomings, the section on error propagation should be removed. I am also wondering why the authors used the out-of-bag residuals and not the residuals from the actual RF predictions. I have not seen any other studies using out-of-bag residuals for regression-kriging, and the authors should elaborate on their reasons for this choice.

Sixthly, the authors use the hydrological DK-model as a covariate in the random forest model. I am wondering if the training points used in the RF model were also used for calibrating the DK-model. If this is the case, it creates a risk of circular logic, as the covariate contains information on the target variable at the location of the training points.

Seventh, the authors state that the sine model used to estimate extreme events could be replaced by an updated version of the DK-model. While I agree that this would improve the estimate of extreme events, it would also introduce another potential source of circular logic, if the DK-model was still used as a covariate. The approach would therefore need to be implemented with great care in order to avoid this.

Lastly, I would like to comment on the use of the term "validation" for accuracy assessment. This is a general concern with the literature as much as a comment on this manuscript in particular. Oreskes (1998) argues that a quantitative model of a complex natural system cannot be considered as truly "validated" until it is used. For example, a conceptually flawed model can still provide good accuracies. The issue becomes even more relevant for machine learning models, where the parameters represent only patterns in the data, not physical processes. Strictly speaking, a machine-learning model can therefore never be truly valid, although it may be accurate and useful. To emphasize this point, I will mention Fourcade et al. (2018), who accurately mapped species distributions with entirely nonsensical covariates. I will encourage the authors

to consider these points when discussing the accuracy of the predictions.

Technical corrections and stylistic suggestions

Generally, the authors refer to "traditional physically-based modelling" several times in the manuscript. I think "conventional" would be more adequate than "traditional", as science has conventions, not traditions. Tradition is a cultural phenomenon. Indeed, in most cases both "conventional" and "traditional" are redundant, as "physically-based modelling" accurately describes what the authors refer to, without any further need of clarification.

Page 2:

L5: "There exists a broad relevancy of the shallow groundwater" –> "The shallow groundwater has a broad relevance"

L9 – L10: "energy partitioning" –> "energy balance"

L13: "The shallow groundwater is also of importance in the urban context" –> "The shallow groundwater is also important in urban contexts"

L19: "a 100 year event with respect to today's average conditions" –> "a 100-year event relative to present average conditions"

L21: "high permeable" –> "highly permeable"

L28: "which hinders to conduct thorough calibration, sensitivity and uncertainty analysis at high resolution" –> "which hinders thorough calibration, and sensitivity and uncertainty analyses at high resolution"

L29: "Further, there exists a general difficulty to parameterize subsurface processes regardless the scale" –> "Furthermore, it is difficult to parameterize subsurface processes regardless of the scale"

Page 3: L3: "Hydrology" –> "hydrology"

L16: "mode" –> "model"

L16: "Before machine learning techniques can build the toolbox of future's environmental decision making" –> "Before machine learning techniques can be considered as a toolbox for environmental decision making"

L25: "Opposed" –> "However"

L29: "or" –> "and"

Page 4:

L3: "The study area encompasses a large part of the Danish peninsular, which is located in Northern Europe (54.5–57.8°N and 8.0–10.9°E) and referred to as Jutland." –> "The study area encompasses a large part of the Jutland peninsula, located in Denmark in northern Europe (54.5–57.8°N; 8.0–10.9°E)."

L5: "the sequence" –> "a sequence"

L6 – L8: The clay contents in eastern Denmark are not very high (10 – 20% for the topsoil). They are higher than the clay contents in western Denmark, but not relative to other areas in the world. It would be more accurate to say that the texture is loamy or that the clay contents are moderately high.

L8: "Weichselian sandy outwash plains" –> "sandy Weichselian outwash plains"

Page 5:

L6: "well data [. . .] was" –> "well data [. . .] were"

Page 6:

L6: "coast" –> "the coastline". This should be the case throughout the manuscript. Also "coastline" → "the coastline".

Page 8:

Table 1: Lowland classification and landscape typology should refer to Madsen et al. (1992).

Table 1: "Drain probability" –> "Probability of artificial drainage"; "Drain class" –> "Soil drainage class".

Page 9:

L13: Bootstrap samples on average contain 63.2% of the data, not 66%.

L25: "The concept of covariate permutation allows to assess the importance of each covariate" –> "Covariate permutation allows an assessment of the importance of each covariate"

Page 12:

L20: "visual" –> "visible"

Page 13:

L2 – L3: Delete "was evident".

Page 17:

L21: "clear a shortcoming" –> "a clear shortcoming"

Page 19:

L3: "that region" –> "the study area"

Page 20:

L14: "allows to model" –> "enables"

References

Fourcade, Y., Besnard, A.G. and Secondi, J., 2018. Paintings predict the distribution of species, or the challenge of selecting environmental predictors and evaluation statistics. Glob. Ecol. Biogeogr. 27(2), 245-256. http://dx.doi.org/10.1111/geb.12684.

Madsen, H.B., Nørr, A.H. and Holst, K.A., 1992. The Danish soil classification. The Royal Danish Geographical Society, Copenhagen, Denmark.

Muscarella, R., Galante, P.J., Soley-Guardia, M., Boria, R.A., Kass, J.M., Uriarte, M., Anderson, R.P. and McPherson, J., 2014. ENMeval: An R package for conducting spatially independent evaluations and estimating optimal model complexity for Maxent ecological niche models. Methods Ecol. Evol. 5(11), 1198-1205. http://dx.doi.org/10.1111/2041-210x.12261.

Oreskes, N., 1998. Evaluation (not validation) of quantitative models. Environ. Health Perspect. 106(Suppl 6), 1453-1460. http://dx.doi.org/10.1289/ehp.98106s61453.

Rudiyanto, Minasny, B., Setiawan, B.I., Saptomo, S.K. and McBratney, A.B., 2018. Open digital mapping as a cost-effective method for mapping peat thickness and assessing the carbon stock of tropical peatlands. Geoderma 313, 25-40. http://dx.doi.org/10.1016/j.geoderma.2017.10.018.
* * *

---

## Referee Comment (RC2) · Katherine Ransom (Referee) · 24 Jun 2019

General Comments

Overall this paper is well written, the methods are scientifically sound, and the work provides a substantial contribution to the current body of knowledge. The sensitivity analysis to provide local variable importance is highly useful and I am not aware of any other studies that provide such a map. This paper is suitable for publication in HESS. I have several comments, detailed below, that relate mainly to the methods descriptions that the authors can address mostly by providing more clarity or discussion related to the specific concerns.

Specific Comments

In the data section, it is stated that 1,900 additional data points were used in the training dataset to represent areas where depth to groundwater is 0. However, later on, namely Figure 1 caption and in the Results section, it is unclear if the 15,000 additional points were used or if it was still just the 1,900. The data section states the data density of the additional points is the same as that of the measured data but this can't be the case if the authors only used 1,900 additional points. Please clarify throughout the text.

In Section 2.2 how is the vertical distance to the nearest water body measured? Are the depth to water measurements involved in this calculation?

Section 2.4 might be more appropriately labeled "Covariate Importance" or "Random Forest Sensitivity to Covariates"

I agree with the previous referee that the RMSE metric is probably better than R2 to quantify the covariate importance in the sensitivity analysis. Please discuss the reason to use R2 and the possibility to recalculate the sensitivity using RMSE.

It is unclear what the authors are referring to in Section 2.4 when they say "each simulation grid". Do they mean each grid cell? The authors state: "prediction is repeated n times until the mean difference across n permutations converges for each simulation grid." Do they mean the mean difference for each grid cell or the mean difference among all grid cells? Please clarify throughout the text.

Section 2.6 should include a description of the software used to calculate the QRFs. Was a special Python package available or was it programmed by the authors following the methods in Meinshausen, 2006?

Section 2.6. This section seems incomplete. Please provide discussion on why the approach can be used if the underlying assumption of no covariance is violated and/or why the approach was used here. What is the purpose of the error propagation/how did the authors use it here? The explanation is provided on page 16 lines 10-14, but should be provided in the methods.

[Figure]

In section 3.1 Random Forest Model, the authors state that "After initial testing, the RF model was parametrized as follows; the number of decision trees was set to 1,000, bootstrapping with replacement was applied to sample the training data, 33% of the covariates were considered to identify the optimal data split" and I am curious what the initial testing entailed and if the authors performed any tuning of these parameters, such as with a cross validation. It could be useful for the authors to more thoroughly describe the process and metrics used for selecting the number of trees and the percent of covariates selected for each tree. This description might also be more appropriate in the methods section.

In section 3.1 Random Forest Model, the authors state that "The oob prediction can be considered as an independent validation test" and the authors did elaborate on this at the end of section 2.3. But readers may benefit from a reminder here that the contribution to the overall oob error from each observation is calculated based upon only the trees which did not contain that specific observation in the bootstrap and provide the reference (Breiman, 2001?). Though, I am not sure if I agree that the oob error can be used as an independent assessment of the generalization/validation error if this is what the authors meant. When predictions are made to unsampled areas or to unseen data, all 1000 trees are used. However, if the above is correct, the oob error is calculated for each observation based upon only a subset of the 1000 trees (n = 340), so the entire model is not assessed when calculating the oob error. The authors might want to consider calculating the testing error to a separate validation/testing set and comparing it to the oob error or providing more discussion on why the oob error also adequately quantifies the generalization error. Additionally, was the coefficient of determination a Pearson correlation coefficient or Nash-Sutcliffe? From the text I gather it is a Nash-Sutcliffe, this should be specified in the text.

Please provide summary statistics for the training data so readers can better understand the reported oob MAE and RMSE.

In section 3.1 and Figure 3, are the very shallow water table points which were consistently over-predicted the same additional points that were added (with 0 depth to water)?

Section 3.2 discusses the results of the prediction sensitivity analysis. From Figure 6 it does appear that this analysis was done on the grid cell level but please clarify in the text (see above).

Section 3.3 should describe why all data including data not in the model was used for RFRK.

From Figure 8 it is hard to tell if there is any variation among grid cells not located at a surface water location. Could the color scale be adjusted to better display the local variation for the RFRK?

Section 4.1. Did the authors compare model results with and without the additional data points of 0 depth to water? If such a scenario was tested it might be useful to discuss here.

Section 4.2 Line 19-23 Were the covariates with low importance expected to be important relative to the covariates ranked as highly important? In addition to the possibilities the authors discuss, the drainage characteristics and topographic wetness index may also be overshadowed by the highly ranked covariates and could become important if the other covariates were removed from the model. If the RF model is not selecting the drainage characteristics and topographic wetness index covariates for splits very often or if splits on these variables occur far down in the trees (near the leaves) then we would not expect the permutations to be highly impactful. Along these lines, did the authors consider calculating other forms of variable importance such as relative importance based on reduction of RMSE attributed to each covariate within the model?

Technical Corrections

Table 1, Column 2, Row 9: "and" instead of "an"?

Figure 5 should have more descriptive labels for covariates, like Table 1.

[Figure]

Page 16 Line 8: do the authors mean each grid cell?

Page 17 Line 22: incomplete sentence?

Page 18 Line 11: "located" instead of "placed"

––––––––––––––––––––––––

---

## Author Comment (AC1) · 22 Jul 2019

Manuscript **hess-2019-212**: "Modelling of the shallow water table at high spatial resolution using Random Forests."

Correspondence to Julian Koch (juko@geus.dk)

Author response to Anders Bjørn Møller. Reviewer evaluation in italic. Author reply in blue font.

*General comments*
*I have read the manuscript "Modelling of the shallow water table at high spatial resolution using Random Forests" submitted to HESS by Koch et al., in order to provide a referee comment.*
*The manuscript is well structured, clear, concise and well written. It addresses the depth to the shallow water table, which is a highly relevant issue, and introduces a number of novel methods in doing so. Some parts of the introduced methods have great potential, not only for hydrological applications but for spatial predictions with machine learning in general.*
*My main concerns with the manuscript lie with some of the specific choices that the authors make in implementing the methods, especially related to the assessment of the accuracy and uncertainties of the predictions. I will elaborate on these concerns in the following section. However, given that the authors address them, the manuscript is highly suitable for publication in HESS.*

**Reply:** We would like to thank Anders Bjørn Møller for his comprehensive review, which raises very thoughtful comments on our manuscript. We are very pleased to have received an overall positive evaluation of our manuscript and will gladly revise it following his comments to further strengthen the scientific quality of our work. We hope that the suggested changes, which are outlined in the response below, will be well received and that the re-submission will be regarded a significant contribution to HESS.

*Specific comments*
*Firstly, I am wondering why the authors choose to map the depth to the shallow water table rather than the elevation of the shallow water table. I would expect the elevation of the shallow water table to show less spatial variation than the depth from the surface. It should therefore be easier to predict, all other things equal. I am sure the authors have good reasons for this choice, but I would like to see them stated explicitly.*

**Reply:** The variable required by the stakeholders is the depth to the groundwater table, which is straightforward to interpret in the context of infrastructure planning and implementing of climate change adaption strategies. This being said, we could have decided to apply the RF model to simulate groundwater elevation which can easily be converted to depth by subtracting it from the surface elevation. In general, we agree with the point raised by the reviewer that the groundwater elevation is more homogenous than its depth. This will be especially the case for shallow sandy aquifers, but in more complex geological settings, such as glacial tills, this is not necessarily the case. Here, a secondary shallow water table often follows the surface elevation. Simulating groundwater elevation instead of depth would be largely driven by the surface elevation and the complex influence of soil and topographical features may diminish. In order to test this assumption, we trained a RF model to simulate groundwater elevation, which we then converted to groundwater depth. The oob prediction is used to assess the accuracy and can be compared to the original RF model from the manuscript that directly simulates the depth. The results are presented in figure 1. The RF model trained against groundwater elevation shows more scatter/deviation, which is also underlined by the statistics. The accuracy of the oob prediction of the original RF model was: $R^2$=0.56, RMSE=1.13 m and MAE=0.76 m. The groundwater elevation RF model was: $R^2$=0.43, RMSE=1.28 m and MAE=0.80 m. The

scores of all three metrics worsened, which strengthens our original decision to simulate the groundwater depth. This comparison will be mentioned in the revised discussion section.

[Figure]

Figure 1 The oob prediction of the original RF from the manuscript, trained against groundwater depth, is compared with an alternative RF model, trained against groundwater elevation. The oob prediction of the latter was converted groundwater depth.

*Secondly, I would like to comment on the use of a sine function to model an annual minimum event. I think it is a useful and generally robust way to address the issue of working with limited data. However, the method could be improved upon in a number of ways. Firstly, the maximum of the curve does not match the maximal observed water levels. The authors could therefore have calculated the uncertainty related to the sine model and, ideally, used these uncertainties in the Random Forest model. The authors already state this in the manuscript, but my second comment is related to the same issue. For training locations with sparse data, the authors used the maximum of the sine curve, but for training locations with more observations, the authors used observed maximum water levels. This choice muddles the results, both in terms of the predicted values and their accuracies. Is it a map of the expected minimum depth to the shallow water table, averaged over a number of years? Or is it a map of an extreme event, observed only in some years? The mixture of training data makes this question difficult to answer.*

**Reply:** The sinusoidal correction model was a necessary step to be able to fully capitalize all available observations of the shallow groundwater system. The training dataset comprises 14916 wells of which 392 have a long timeseries (more than 5 observations). In other words, 97% of the training data underwent a correction with the defined sine models. The muddling of the results that the reviewer refers to is therefore not that severe as 97% of the data are processed in the same way and should represent a minimum event that is expected to occur every year. The minimum depth at the remaining 3% may slightly disagree, but we excluded very dry years from our analysis (1992-1997 and 2018) that could potentially lead to large disagreements. For the two examples in Figure 2 from the manuscript, the observed minimum values match quite well the sine model, with a deviation of approximately 10-20 cm. This falls within the uncertainty of the measurement itself. Therefore, we believe that the applied sine correction is a robust and appropriate approach. We acknowledge the uncertainties related to this processing step. The variability at the 392 wells with long timeseries has been investigated to infer the variability (amplitude) of the sine models for 27 different hydrogeological units. Expert knowledge was supplemented for groups that were poorly informed; i.e. only few available wells. The defined amplitudes are uncertain as the 27 groups may represent a crude classification and some groups are only represented by a limited number of wells. As we already outline in the paper, a more physically-based correction is needed for future applications. Using a hydrological model

for the correction would allow us to differentiate between inter-annual variations of the groundwater amplitude at grid level. Also, we could differentiate between dry and wet years.

We carried out a test to address the reviewers comments. We trained a RF model against the 97% of the observations that underwent the sine correction and withheld the 3% with more than 5 measurements to test the model. We then compared these results with the oob prediction of the original model. As indicated by the figure below, the predictions match quite well with a few exceptions.

[Figure]

Figure 2 The plot shows predictions only for the 392 wells with long timeseries (more than 5 observations). The x axis presents the oob prediction of the original RF model from the manuscript. On the y axis, predictions of the 392 wells based on a RF model trained against the remaining 97% of the data with less than 6 observations.

Below, the same data is plotted against the observations. The two RF models differ, but the fact that no systematic difference is present strengthens our argument to use both, sine corrected observations and true observations, in the modeling process.

[Figure]

Figure 3 Same data as Figure 2, but plotted against the observations.

*Thirdly, I have concerns about the way that the authors assess the accuracy of the predictions. The training dataset shows a high degree of clustering. Therefore, when the authors use the out-of-bag predictions for assessing the accuracy, the points used for*

*assessing the accuracy will be located close to the training points used for making the predictions. It is very likely that the values are spatially autocorrelated, and the stated accuracy is therefore probably not representative for the study area as a whole. I would expect the accuracy to be lower for the parts of the study area that do not have a high density of observations. A spatially structured accuracy assessment, as proposed for example by Muscarella et al. (2014), would most likely provide a more representative accuracy assessment. Furthermore, I am wondering if the authors used all the training points for the predictions. The training dataset contained both groundwater and surface water observations. However, the aim is not to predict surface water levels, and I would therefore say that one could justify removing the surface water points from the out-of-bag predictions when assessing the accuracy.*

**Reply:** Inspired by this comment we prepared a 10-fold cross validation test to supplement the reported oob accuracy. Here we applied a standard method to partition the data randomly without considering the locations of the samples, as proposed by Muscarella et al. (2014). We agree that some of our data is clustered around cities or infrastructure projects. However, given a spatial autocorrelation of the RF residuals of 200 m (see variogram figure in the manuscript), we believe that with a simulation resolution of 50 m, we do not have to consider the spatial autocorrelation of the data for the partitioning for the 10-fold validation test.

We agree with the reviewer that the additional surface water observations with a groundwater depth of 0 m should not be included in the accuracy assessment. This was already considered in the submitted mansucript and will be stated more clearly in the revised version.

For the 10-fold cross validation test, the dataset was randomly split in 10 sets of approximately the same size. Then 10 RF models were trained on 90% of the data so that each set was left out once and could be used for validation purposes. The results are strikingly similar as compared to the oob prediction as shown in figure 4. Figure 5 depicts a scatterplot of the ~17k training samples comparing the predictions form the oob approach and the cross validation. In our opinion, the agreement is convincing which qualifies the oob prediction as an appropriate way to quantify the generalization error of our RF model. Furthermore, the statistics were very similar as well, as reported in the table below. We believe that this table and the addition of the 10-fold cv test is valuable for the revised manuscript and it will therefore be added to the re-submission. Please be aware that the oob metric scores vary slightly compared to the ones mentioned in the original manuscript. We regret that the script used to compute the scores did not read the latest data.

|  | $R^2$ | RMSE m | MAE m |
|---|---|---|---|
| oob | 0.56 | 1.13 | 0.76 |
| 10-fold cv | 0.55 | 1.15 | 0.77 |

[Figure]

Figure 4 Comparison of oob prediction and 10-fold cross validation against observations.

[Figure]

Figure 5 Direct comparison of oob prediction and 10-fold cross validation.

*Fourthly, I very much like the way that the authors handle covariate importance. Being able to assess covariate importance in geographic space is potentially extremely useful, for both researchers and end users. However, I do not think that decrease in R2 is the best measure of covariate importance. One can potentially obtain a high R2, even if the absolute values are inaccurate. A better choice would therefore be to assess the relative change in a measure that accounts for absolute values, such as RMSE, Lin's concordance or the Nash-Sutcliffe efficiency.*

**Reply:** We appreciate the reviewer's thoughts on our spatial covariate importance analysis. We need to clarify that the analysis on the prediction dataset which allows us to map covariate importance in space, uses the absolute difference between the permuted prediction and the original prediction. In this way, the reviewer's concerns were already considered in the original submission. For the purpose of the review we tested if the squred differences could give another result of the spatial covariate importance. This was not the case and the differences to the original results were minor. Therefor we decided not to include the results here or in the revised manuscript.

The decrease in $R^2$ was used to quantify covariate importance for the training dataset. Here we completely agree with the reviewer that the applied metric may not be the most suitable one. Therefore we tested the

analysis with the RMSE instead and results are shown in the figures below. The two figures below show the results based on the original assessment of the R² and the newly tests RMSE assessment. Although the percentages vary between the two metrics, the same conclusions in terms of relative covariate importance can be drawn. Therefore, we decided not to add the RMSE based figure in the revised manuscript, but instead, we will mention that a sensitivity analysis based on RMSE was made and that it yielded the same conclusions in terms of covariate sensitivity ranking.

[Figure]

Figure 6 Covariate importance based on R2. Same as Figure 5 in the manuscript.

[Figure]

Figure 7 Covariate importance based on RMSE

*Fifthly, while I appreciate that the authors assessed the uncertainties of the predictions in two different ways, I do not think that combining them is justified. The theoretical basis for the approach seems scarce. Both the RF uncertainties and the residuals used for kriging relate to the same model, and it is therefore a stretch to say that they are independent. Furthermore, quantile regression forest should be able to assess uncertainties quite accurately without any further elaboration, as shown for example by Rudiyanto et al. (2018). I think a large part of the spatial autocorrelation in the residuals would disappear, if one takes into account the uncertainties related to the RF predictions. The uncertainties in the predictions make the residuals uncertain as well, which complicates regression-kriging. When experimenting with techniques, as the authors do, it is important to set aside an independent part of the dataset to be able to assess the accuracy of the estimated uncertainties. However, the authors do not do this, and it is therefore impossible to assess if the error propagation actually leads to a better estimate of the uncertainties. Unless the authors can adequately adress these shortcomings, the section on error propagation should be removed. I am also*

*wondering why the authors used the out-of-bag residuals and not the residuals from the actual RF predictions. I have not seen any other studies using out-of-bag residuals for regression-kriging, and the authors should elaborate on their reasons for this choice.*

**Reply:** We acknowledge that the uncertainty propagation of RFRK and QRF was quite a leap given the current analysis in our manuscript. More work will be required to test the underlying assumption that the uncertainty sources have no significant covariance and thus can be combined. We have decided to remove section 2.7 and related text from the discussion. Figure 8 will be updated as well. However, we do believe it is a valuable contribution to present, compare and discuss the RFRK and QRF based uncertainty estimations.

It is correct that we have used the oob predictions for the residual kriging and that this does not seem to be common practice in the literature. The RF model we apply is fully expanded until each leaf contains only a single data point. In this way, the standard RF prediction will be very close to the actual observation, as the observation value will be included in ~63% of the trees. This would lead to much lower residuals that do not really represent the generalization error. In this way, it would make no sense to interpolate the residuals for the standard RF prediction to unsampled locations. This assumption will be mentioned in the method section of the revised manuscript.

*Sixthly, the authors use the hydrological DK-model as a covariate in the random forest model. I am wondering if the training points used in the RF model were also used for calibrating the DK-model. If this is the case, it creates a risk of circular logic, as the covariate contains information on the target variable at the location of the training points.*

**Reply:** This is correct, some of the data was also used for the calibration of the DK model. However, for the purpose of this study, we have collected additional data from various sources (municipalities, region and consultancies) that were not yet in the national database and thus not been used for the calibration of the DK model. Further the shallow observations have so far received a low weight in the calibration of the DK model. Therefore, we do not believe that this problem will be a limitation of using the DK model as covariate in our model.

*Seventh, the authors state that the sine model used to estimate extreme events could be replaced by an updated version of the DK-model. While I agree that this would improve the estimate of extreme events, it would also introduce another potential source of circular logic, if the DK-model was still used as a covariate. The approach would therefore need to be implemented with great care in order to avoid this.*

**Reply:** We still believe that a more physically based correction of the observations is the way to move forward. Nevertheless, we agree with the reviewer that this can potentially lead to some issues of circular logic. We decided to remove this outlook from the manuscript as this is not really related to any of the results presented and thorough testing would be required before such a method could be implemented.

*Lastly, I would like to comment on the use of the term "validation" for accuracy assessment. This is a general concern with the literature as much as a comment on this manuscript in particular. Oreskes (1998) argues that a quantitative model of a complex natural system cannot be considered as truly "validated" until it is used. For example, a conceptually flawed model can still provide good accuracies. The issue becomes even more relevant for machine learning models, where the parameters represent only patterns in the data, not physical processes. Strictly speaking, a machine-learning model can therefore never be truly valid, although it may be accurate and useful. To emphasize this point, I will mention Fourcade et al. (2018), who accurately mapped species distributions with entirely nonsensical covariates. I will encourage the authors to consider these points when discussing the accuracy of the predictions.*

**Reply:** We are very much aware of the discussion on the term validation initiated by Oreskes (1998). In the revised manuscript we will downtown the term validation and use terms like "accuracy assessment" or "model evaluation" instead. However, we believe it is beyond the scope of this paper to discuss if machine learning models can be considered valid after being successfully evaluated against independent observations.

*Technical corrections and stylistic suggestions*

**Reply:** We appreciate this thorough technical/editorial review. We agree to all points raised by A.B. Møller and will updated the revised manuscript accordingly.

*Generally, the authors refer to "traditional physically-based modelling" several times in the manuscript. I think "conventional" would be more adequate than "traditional", as science has conventions, not traditions. Tradition is a cultural phenomenon. Indeed, in most cases both "conventional" and "traditional" are redundant, as "physically-based modelling" accurately describes what the authors refer to, without any further need of clarification.*
*Page 2:*
*L5: "There exists a broad relevancy of the shallow groundwater" –> "The shallow groundwater has a broad relevance"*
*L9 – L10: "energy partitioning" –> "energy balance"*
*L13: "The shallow groundwater is also of importance in the urban context" –> "The shallow groundwater is also important in urban contexts"*
*L19: "a 100 year event with respect to today's average conditions" –> "a 100-year event relative to present average conditions"*
*L21: "high permeable" –> "highly permeable"*
*L28: "which hinders to conduct thorough calibration, sensitivity and uncertainty analysis at high resolution" –> "which hinders thorough calibration, and sensitivity and uncertainty analyses at high resolution"*
*L29: "Further, there exists a general difficulty to parameterize subsurface processes regardless the scale" –> "Furthermore, it is difficult to parameterize subsurface processes regardless of the scale"*
*Page 3: L3: "Hydrology" –> "hydrology"*
*L16: "mode" –> "model"*
*L16: "Before machine learning techniques can build the toolbox of future's environmental decision making" –> "Before machine learning techniques can be considered as a toolbox for environmental decision making"*
*L25: "Opposed" –> "However"*
*L29: "or" –> "and"*
*Page 4:*
*L3: "The study area encompasses a large part of the Danish peninsular, which is located in Northern Europe (54.5–57.8_N and 8.0–10.9_E) and referred to as Jutland." –> "The study area encompasses a large part of the Jutland peninsula, located in Denmark in northern Europe (54.5–57.8_N; 8.0–10.9_E)."*
*L5: "the sequence" –> "a sequence"*
*L6 – L8: The clay contents in eastern Denmark are not very high (10 – 20% for the topsoil). They are higher than the clay contents in western Denmark, but not relative to other areas in the world. It would be more accurate to say that the texture is loamy or that the clay contents are moderately high.*
*L8: "Weichselian sandy outwash plains" –> "sandy Weichselian outwash plains"*
*Page 5:*
*L6: "well data [. . .] was" –> "well data [. . .] were"*
*Page 6:*

*L6: "coast" –> "the coastline". This should be the case throughout the manuscript. Also "coastline" ! "the coastline".*

*Page 8:*

*Table 1: Lowland classification and landscape typology should refer to Madsen et al. (1992).*

*Table 1: "Drain probability" –> "Probability of artificial drainage"; "Drain class" –> "Soil drainage class".*

*Page 9:*

*L13: Bootstrap samples on average contain 63.2% of the data, not 66%.*

*L25: "The concept of covariate permutation allows to assess the importance of each covariate" –> "Covariate permutation allows an assessment of the importance of each covariate"*

*Page 12:*

*L20: "visual" –> "visible"*

*Page 13:*

*L2 – L3: Delete "was evident".*

*Page 17:*

*L21: "clear a shortcoming" –> "a clear shortcoming"*

*Page 19:*

*L3: "that region" –> "the study area"*

*Page 20:*

*L14: "allows to model" –> "enables"*

---

## Author Comment (AC2) · 22 Jul 2019

Manuscript **hess-2019-212**: "Modelling of the shallow water table at high spatial resolution using Random Forests."

Correspondence to Julian Koch (juko@geus.dk)

Author response to Katherine Ransom. Reviewer evaluation in italic. Author reply in blue font.

*General Comments*
*Overall this paper is well written, the methods are scientifically sound, and the work provides a substantial contribution to the current body of knowledge. The sensitivity analysis to provide local variable importance is highly useful and I am not aware of any other studies that provide such a map. This paper is suitable for publication in HESS. I have several comments, detailed below, that relate mainly to the methods descriptions that the authors can address mostly by providing more clarity or discussion related to the specific concerns.*

**Reply:** We would like to thank Katherine Ransom for her thorough revision of our manuscript. We are very pleased that our modelling approach of the shallow groundwater system was generally well received. The comments made by Katherine Ransom raise valuable points and a rigorous revision following her suggestions will certainly improve the scientific quality of our work. We hope that the suggested changes will be appreciated and that the re-submitted manuscript will be rated fit for publication in HESS.

*Specific Comments*
*In the data section, it is stated that 1,900 additional data points were used in the training dataset to represent areas where depth to groundwater is 0. However, later on, namely Figure 1 caption and in the Results section, it is unclear if the 15,000 additional points were used or if it was still just the 1,900. The data section states the data density of the additional points is the same as that of the measured data but this can't be the case if the authors only used 1,900 additional points. Please clarify throughout the text.*

**Reply:** A total of 15k additional observations were placed along streams, coastline and in lakes. Our intention was to constrain the RF model with critical information that was otherwise not provided by the wells alone. We realized that including all 15k in the training would negatively affect the RF prediction, as the model was strongly biased to depth 0 m. Therefore we decided to use only a subset of the additional observations. We found 1900 a suitable number, because this reflects the same well density (1 well per km$^2$) as found in the well dataset. For the additional observations, the density refers to the area of surface water (stream, lakes and coastline) in 50 m grid resolution. In this way, the amount of wells and additional observations was balanced. However, in the residual kriging we used all 15k wells to correct the water table at locations with surface water where we expect a depth of 0 m. This will be stated more precisely in the revised manuscript and the caption of figure 1 will be changed accordingly.

*In Section 2.2 how is the vertical distance to the nearest water body measured? Are the depth to water measurements involved in this calculation?*

**Reply:** The vertical distance to the nearest waterbody is only in relation to surface waterbodies. For a given grid, we first find the nearest waterbody (lake, river, coastline). Then we compute the elevation difference of the given grid and the closest waterbody grid. This is will be stated more clearly in the revised manuscript.

*Section 2.4 might be more appropriately labeled "Covariate Importance" or "Random Forest Sensitivity to Covariates"*

**Reply:** We agree the term "Covariate Importance" would be more fitting to the standard RF terminology. However, the hydrological community may relate more to the term "Sensitivity", but readers may think of

model parameter sensitivity which is not what we are addressing here. We will label the section "Covariate Sensitivity" instead.

*I agree with the previous referee that the RMSE metric is probably better than R2 to quantify the covariate importance in the sensitivity analysis. Please discuss the reason to use R2 and the possibility to recalculate the sensitivity using RMSE.*

**Reply:** We completely agree to the relevance of this point. In order to address this issue we computed covariate importance based on the relative increase in RMSE. The two figures below show the results based on the original assessment of the $R^2$ and the newly RMSE assessment. Although the percentages vary between the two metrics, the same conclusions in terms of relative covariate importance can be drawn. Therefore, we decided not to add the RMSE based figure in the revised manuscript, but instead, we will mention that we have conducted the sensitivity analysis based on RMSE and that it yielded the same conclusions in terms of covariate sensitivity ranking.

[Figure]

Figure 1 Covariate importance based on R2. Same as Figure 5 in the manuscript.

[Figure]

Figure 2 Covariate importance based on RMSE

In terms of the spatial mapping of covariance importance we already use the absolute difference between the original prediction and the permuted predictions. We have tested the squared differences but could not notice a significant change. We will be more explicit in the revised manuscript with respect to what metrics are used to compute the covariance importance and mention our tests of using alternative metrics.

*It is unclear what the authors are referring to in Section 2.4 when they say "each simulation grid". Do they mean each grid cell? The authors state: "prediction is repeated n times until the mean difference across n permutations converges for each simulation*

*grid." Do they mean the mean difference for each grid cell or the mean difference
among all grid cells? Please clarify throughout the text.*

**Reply:** Correct, we meant each grid cell. For each grid cell, the difference to the original prediction is recorded for n permutations. We then compute the cumulative mean across these permutations and check if the mean converges.

*Section 2.6 should include a description of the software used to calculate the QRFs.
Was a special Python package available or was it programmed by the authors following
the methods in Meinshausen, 2006?*

**Reply:** We have used the scikit learn implementation of RF in python for our modelling work. To our knowledge, QRF is not implemented in scikit learn yet. Therefore, we have built our own QRF implementation as a workaround using the functions provided by scikit learn.

*Section 2.6. This section seems incomplete. Please provide discussion on why the
approach can be used if the underlying assumption of no covariance is violated and/or
why the approach was used here. What is the purpose of the error propagation/how
did the authors use it here? The explanation is provided on page 16 lines 10-14, but
should be provided in the methods.*

**Reply:** We acknowledge that the uncertainty propagation of RFRK and QRF was quite a leap given the current analysis in our manuscript. More work will be required to test the underlying assumption that the uncertainty sources have no significant covariance and thus can be combined. We have decided to remove section 2.7 and related text from the discussion. Figure 8 will be updated as well. However, we do believe it is a valuable contribution to present, compare and discuss the RFRK and QRF based uncertainty estimations.

*In section 3.1 Random Forest Model, the authors state that "After initial testing, the
RF model was parametrized as follows; the number of decision trees was set to 1,000,
bootstrapping with replacement was applied to sample the training data, 33% of the
covariates were considered to identify the optimal data split" and I am curious what
the initial testing entailed and if the authors performed any tuning of these parameters,
such as with a cross validation. It could be useful for the authors to more thoroughly describe
the process and metrics used for selecting the number of trees and the percent
of covariates selected for each tree. This description might also be more appropriate
in the methods section.*

**Reply:** With regard to the tuning of the RF hyper-parameters we have assessed two parameters in more detail: the number of tress and the n_leaf parameter, which controls the pruning of the decision trees. This initial test was conducted for a subdomain, which covers approximately 10% of our entire domain. Figure 3 and figure 4 show the RF performance for numerous combinations of n_tree and n_leaf parameters. We concluded that the performance converged for 1000 trees and that the trees should be fully expanded (n_leaf=1). Originally, we did not test the max_feature parameter, which controls how many covariates are selected randomly for optimizing each split. We chose 33%, because a lower number generally decreases computational time and increases diversity among the trees. Both aspects are desirable. For the purpose of this review, we briefly tested the sensitivity of the max_features parameter and observed that the $R^2$ for the oob prediction was affected in the third digit and the MAE (mean absolute error) in the second. Thus we conclude that the max_feature parameters is not sensitive for our application. We do not think that this information is necessarily relevant to readers and will therefore not expand the method description of the revised manuscript.

[Figure]

Figure 3 Initial testing of the RF hyper-parameters n_leaf and n_trees with respect to the R2 metric.

[Figure]

Figure 4 Initial testing of the RF hyper-parameters n_leaf and n_trees with respect to the MAE (mean absolute error) metric.

*In section 3.1 Random Forest Model, the authors state that "The oob prediction can be considered as an independent validation test" and the authors did elaborate on this at the end of section 2.3. But readers may benefit from a reminder here that the contribution to the overall oob error from each observation is calculated based upon only the trees which did not contain that specific observation in the bootstrap and provide the reference (Breiman, 2001?). Though, I am not sure if I agree that the oob error can be used as an independent assessment of the generalization/validation error if this is what the authors meant. When predictions are made to unsampled areas or to unseen data, all 1000 trees are used. However, if the above is correct, the oob error is calculated for each observation based upon only a subset of the 1000 trees (n = 340), so the entire model is not assessed when calculating the oob error. The authors might want to consider calculating the testing error to a separate validation/testing set and comparing it to the oob error or providing more discussion on why the oob error also adequately quantifies the generalization error. Additionally, was the coefficient of determination a Pearson correlation coefficient or Nash-Sutcliffe? From the text I gather it is a Nash-Sutcliffe, this should be specified in the text.*

*Please provide summary statistics for the training data so readers can better understand the reported oob MAE and RMSE.*

**Reply:** As suggested by the reviewer we will add further elaboration on how the oob predication is calculated to the revised manuscript. In order to clarify, it is not correct that only a subset of the trees are validated when using the oob prediction. The idea is that each tree uses its own bootstrap sample for traning.

In that way each tree also has its own oob sample that can be used to validated that specific tree. In the end, we can average over all trees where a sample has been retained as oob to obtain the final oob prediction. In this way, all trees have been validated when using the oob prediction.

We have not used the NSE metric to quantify model performance in the original submission. Instead we have applied the coefficient of determination ($R^2$), mean-absolute-error (MAE) and root-mean-squared-error (RMSE). In the revised manuscript, we will state the evaluation metrics more explicitly, but we will omit equations as these are quite generic metrics that the readers of HESS will be familiar with.

In order to investigate if the oob prediction is a reliable source to quantify the generalizability of a RF model we have conducted a 10-fold cross validation test. For this, the dataset was randomly split in 10 sets of approximately the same size. Then 10 RF models were trained on 90% of the data so that each set was left out once and could be used for validation purposes. The results are strikingly similar as compared to the oob prediction as shown in figure 5. Figure 6 depicts a scatterplot of the ~17k training samples comparing the predictions form the oob approach and the cross validation. In our opinion, the agreement is convincing which qualifies the oob prediction as an appropriate way to quantify the generalization error of our RF model. Furthermore, the statistics were very similar as well, as reported in the table below. We believe that this table and the addition of the 10-fold cv test is valuable for the revised manuscript and it will therefore be added to the re-submission. Please be aware that the oob metric scores vary slightly to the ones mentioned in the original manuscript. We regret that the script used to compute the scores did not read the latest data.

| | $R^2$ | RMSE m | MAE m |
|---|---|---|---|
| oob | 0.56 | 1.13 | 0.76 |
| 10-fold cv | 0.55 | 1.15 | 0.77 |

[Figure]

Figure 5 Comparison of oob prediction and 10-fold cross validation with respect to observations.

[Figure]

Figure 6 Comparison of oob prediction and 10-fold cross validation.

*In section 3.1 and Figure 3, are the very shallow water table points which were consistently over-predicted the same additional points that were added (with 0 depth to water)?*

**Reply:** This is not the case. The systematically overestimated shallow observations are consistently placed in the glacial till. It seems that the RF lacks covariate information to adequately reflect such conditions. Glacial tills can be very complex and at the current stage we do not have the required hydrogeological data with a relevant spatial resolution to resolve this issue. We will further elaborate on this shortcoming in the revised manuscripts.

*Section 3.2 discusses the results of the prediction sensitivity analysis. From Figure 6 it does appear that this analysis was done on the grid cell level but please clarify in the text (see above).*

**Reply:** Correct, the results of the sensitivity analysis that are presented in figure 6 (in the manuscript) are calculated on grid cell level. However, please be aware that we have implemented two approaches to quantify covariate importance. The first is the conventional assessment that applies the concept of permutation accuracy on the training dataset. Results for this analysis are given in Figure 5 (in the manuscript). The results of our novel contribution to assess sensitivity of the predication dataset at grid level are given in Figure 6 (in the manuscript). Both approaches are introduced in section 2.4. We will be more explicit about this distinction in the revised manuscript.

*Section 3.3 should describe why all data including data not in the model was used for RFRK.*

**Reply:** As mentioned earlier, the 15k additional observations were reduced to 1900 in order to be in balance with the well observations. However, this reduction was only affecting the RF training. For the RFRK we chose to include all additional observations to ensure that the final groundwater estimates are close to the surface at locations where surface water is present. This ensures physical consistency. The correlation length of the variogram model used to model the RF residuals is set to 200 m. This limits the effect of the additional observations with a groundwater depth of 0 m to a close vicinity.

*From Figure 8 it is hard to tell if there is any variation among grid cells not located at a surface water location. Could the color scale be adjusted to better display the local variation for the RFRK?*

**Reply:** It is correct that the uncertainty of the RFRK model does not vary at grids further away than 200 m (correlation length of the variogram) from an observations. Beyond 200 m distance the uncertainty will be equal to the sill of the variogram model ($1.02 \text{ m}^2$). In that way changing the color scale would not change the figure. This is a natural characteristics of kriging based interpolations and will be mentioned explicitly in the revised manuscript.

*Section 4.1. Did the authors compare model results with and without the additional data points of 0 depth to water? If such a scenario was tested it might be useful to discuss here.*

**Reply:** We have compiled the figure below (figure 7) to address the effect of adding the additional observations to the RF model. The zoom extent is approximately 5 km from left to right and contains a lake, river system and wetlands. If we leave out the additional observations in the training dataset the final RF prediction does not capture the interaction between surface water and groundwater very well. In Denmark, it can be assumed that all surface waterbodies are connected to the shallow groundwater system. This example should underlie the importance of the additional observations in the applied RF model and will be discussed in further detail in the revised manuscript.

[Figure]

Figure 7 Results from two training scenarios: The top was trained against well observations plus the 1900 additional observation with a groundwater depth of 0 m. The bottom was trained exclusively against the well observations.

*Section 4.2 Line 19-23 Were the covariates with low importance expected to be important relative to the covariates ranked as highly important? In addition to the possibilities the authors discuss, the drainage characteristics and topographic wetness index may also be overshadowed by the highly ranked covariates and could become important if the other covariates were removed from the model. If the RF model is not selecting the drainage characteristics and topographic wetness index covariates for splits very often or if splits on these variables occur far down in the trees (near the leaves) then we would not expect the permutations to be highly impactful. Along these lines, did the authors consider calculating other forms of variable importance such as relative importance based on reduction of RMSE attributed to each covariate within the model?*

**Reply:** In one of our previous replies we showed the results of the covariate importance analysis based on the relative increase in RMSE as an addition to the $R^2$ based assessment and argued why it does not provide any additional insights. The statement by the reviewer is correct and gives a good technical explanation of why some covariates receive a low importance score. We will include some of these thoughts in the revised manuscript.

*Technical Corrections*

**Reply:** We appreciate these technical/editorial suggestions. We agree to all points raised by K. Ransom and will update the revised manuscript accordingly.

*Table 1, Column 2, Row 9: "and" instead of "an"?*
*Figure 5 should have more descriptive labels for covariates, like Table 1.*
*Page 16 Line 8: do the authors mean each grid cell?*
*Page 17 Line 22: incomplete sentence?*
*Page 18 Line 11: "located" instead of "placed"*